# More is Better in Modern Machine Learning:
## when Infinite Overparameterization is Optimal and Overfitting is Obligatory

**James B. Simon**[*]
UC Berkeley & Imbue

**Dhruva Karkada**
UC Berkeley

**Nikhil Ghosh**
UC Berkeley

**Mikhail Belkin**
UC San Diego

## Abstract

In our era of enormous neural networks, empirical progress has been driven by the philosophy that *more is better*. Recent deep learning practice has found repeatedly that larger model size, more data, and more computation (resulting in lower training loss) improves performance. In this paper, we give theoretical backing to these empirical observations by showing that these three properties hold in random feature (RF) regression, a class of models equivalent to shallow networks with only the last layer trained.

Concretely, we first show that the test risk of RF regression decreases monotonically with both the number of features and the number of samples, provided the ridge penalty is tuned optimally. In particular, this implies that infinite width RF architectures are preferable to those of any finite width. We then proceed to demonstrate that, for a large class of tasks characterized by powerlaw eigenstructure, training to near-zero training loss is *obligatory*: near-optimal performance can *only* be achieved when the training error is much smaller than the test error. Grounding our theory in real-world data, we find empirically that standard computer vision tasks with convolutional neural tangent kernels clearly fall into this class. Taken together, our results tell a simple, testable story of the benefits of overparameterization, overfitting, and more data in random feature models.

## 1 Introduction

It is an empirical fact that more is better in modern machine learning. State-of-the-art models are commonly trained with as many parameters and for as many iterations as compute budgets allow, often with little regularization. This ethos of enormous, underregularized models contrasts sharply with the received wisdom of classical statistics, which suggests small, parsimonious models and strong regularization to make training and test losses similar. The development of new theoretical results consistent with the success of overparameterized, underregularized modern machine learning has been a central goal of the field for some years.

How might such theoretical results look? Consider the well-tested observation that wider networks virtually always achieve better performance, so long as they are properly tuned (Kaplan et al., 2020; Hoffmann et al., 2022; Yang et al., 2022). Let $\mathcal{E}_{\text{te}}(n, w, \boldsymbol{\theta})$ denote the expected test error of a network with width $w$ and training hyperparameters $\boldsymbol{\theta}$ when trained on $n$ samples from an arbitrary distribution. A satisfactory explanation for this observation might be a hypothetical theorem which states the following:

$$\text{If } w' > w, \text{ then } \min_{\boldsymbol{\theta}} \mathcal{E}_{\text{te}}(n, w', \boldsymbol{\theta}) < \min_{\boldsymbol{\theta}} \mathcal{E}_{\text{te}}(n, w, \boldsymbol{\theta}).$$

Such a result would do much to bring deep learning theory up to date with practice. In this work, we take a first step towards this general result by proving it in the special case of RF regression — that is, for shallow networks with only the second layer trained. Our Theorem 1 states that, for RF regression, more features (as well as more data) is better, and thus infinite width is best. To our knowledge, this is the first analysis directly showing that for arbitrary tasks, wider is better for networks of a certain architecture.

How might a comparable result for overfitting look? It is by now established wisdom that optimal performance in many domains is achieved when training deep networks to nearly the point of

---

[*]james.simon@berkeley.edu

interpolation (i.e. zero training error), with train error many times smaller than test error ( Neyshabur et al. (2015); Zhang et al. (2017); Belkin et al. (2018), Appendix F of Hui & Belkin (2021)). However, unlike the statement "wider is better," the statement "interpolation is optimal" cannot be true for generic task distributions: we can see this a priori by noting that any fitting at all can only harm us if the task is pure noise and the optimal predictor is thus zero. Indeed, Nakkiran & Bansal (2020) and Mallinar et al. (2022) empirically observe that training to interpolation harms the performance of real deep networks on sufficiently noisy tasks. This suggests that we instead ought to seek an appropriate class $\mathcal{C}$ of model-task pairs — ideally general enough to include realistic tasks — such that a hypothetical statement of the following form is true:

$$\text{For model-task pairs in } \mathcal{C}, \text{ at optimal regularization, it holds that } \mathcal{R}_{\text{tr/te}} := \frac{\mathcal{E}_{\text{tr}}}{\mathcal{E}_{\text{te}}} \ll 1.$$

Here we have defined the *fitting ratio* $\mathcal{R}_{\text{tr/te}}$ to be the ratio of train error $\mathcal{E}_{\text{tr}}$ and test error $\mathcal{E}_{\text{te}}$ and we have suppressed the arguments of $\mathcal{E}_{\text{tr}}$ and $\mathcal{E}_{\text{te}}$ for the sake of generality. We take a step towards this general result, too, by proving a sharp statement to this effect for kernel ridge regression (KRR), including infinite-feature RF regression and infinite-width neural networks of any depth in the kernel regime (Jacot et al., 2018; Lee et al., 2019). Letting $\mathcal{C}$ be the set of tasks with *powerlaw eigenstructure* (Definition 2), our Theorem 2 states that under mild conditions on the powerlaw exponents, not only is $\mathcal{R}_{\text{tr/te}} \ll 1$ at optimal regularization, but in fact this overfitting is *obligatory*: attaining near-optimal test error *requires* that $\mathcal{R}_{\text{tr/te}} \ll 1$.[1] Crucially, we put our proposed explanation to the experimental test: we clearly find that the eigenstructure of standard computer vision tasks with convolutional neural kernels displays powerlaw decay in satisfaction of our "obligatory overfitting" criteria, and indeed optimality occurs at $\mathcal{R}_{\text{tr/te}} \approx 0$ for these tasks (Figure 2).

All our main results rely on closed form estimates for the train and test error of RF regression and KRR in terms of task eigenstructure. We derive such an estimate for RF regression, and our "more is better" conclusion (Theorem 1) follows quickly from this general result. This estimate relies on a Gaussian universality ansatz (which we validate empirically) and becomes exact in an appropriate asymptotic limit, though we see excellent agreement with experiment even at modest size. When we study overfitting in KRR, which is the infinite-feature limit of RF regression, we use the infinite-feature limit of our eigenframework, which recovers a well-known risk estimate for (kernel) ridge regression extensively investigated in the recent literature (Sollich, 2001; Bordelon et al., 2020; Jacot et al., 2020a; Dobriban & Wager, 2018; Hastie et al., 2022). We solve this eigenframework for powerlaw task eigenstructure, obtaining an expression for test error in terms of the powerlaw exponents and the fitting ratio $\mathcal{R}_{\text{tr/te}}$ (Lemma 10), and our "obligatory overfitting" conclusion (Theorem 2 and Corollary 1) follows from this general result. Remarkably, we find that real datasets match our proposed powerlaw structure so well that we can closely predict test error as a function of $\mathcal{R}_{\text{tr/te}}$ purely from experimentally extracted values of the powerlaw exponents $\alpha, \beta$ (Figure 2). To conclude, we return to the question of how one ought to view modern machine learning, suggesting some intuitions consistent with our findings.

Concretely, our contributions are as follows:

- We obtain general closed-form estimates for the train and test risk of RF regression in terms of task eigenstructure (Section 4.1).

- We conclude from the general estimate for test risk that, at optimal ridge parameter, more features and more data are strictly beneficial (Theorem 1).

- We study KRR for tasks with powerlaw eigenstructure, finding that for a subset of such tasks, overfitting is obligatory: optimal performance is only achieved at small or zero regularization (Theorem 2).

- We demonstrate that standard image datasets with convolutional kernels satisfy our criteria for obligatory overfitting (Figure 2).

---

[1]This is in contrast with the proposed phenomenon of "benign overfitting" (Bartlett et al., 2020; Tsigler & Bartlett, 2023) in which interpolation (i.e. training to zero train loss) is merely harmless, incurring only a sub-leading-order cost relative to optimal regularization. In our "obligatory overfitting" regime, interpolation is *necessary*, and *not* interpolating incurs a *leading-order* cost.

## 2    RELATED WORK

**The benefits of overparameterization.** Much theoretical work has aimed to explain the benefits of overparameterization. Belkin et al. (2019) identify a "double-descent" phenomenon in which, for certain underregularized learning rules, increasing overparameterization improves performance. Ghosh & Belkin (2023) show that only highly overparameterized models can both interpolate noisy data and generalize well. Roberts et al. (2022); Atanasov et al. (2023); Bordelon & Pehlevan (2023) show that neural networks of finite width can be viewed as (biased) noisy approximations to their infinite-width counterparts, with the noise decreasing as width grows, which is consistent with our conclusion that wider is better for RF regression. Nakkiran et al. (2021) prove that more features benefits RF regression in the special case of isotropic covariates; our Theorem 1 extends their results to the general case, resolving a conjecture of theirs. Concurrent work (Patil & Du, 2023) also resolves this conjecture, showing sample-wise monotonicity for ridge regression. Yang & Suzuki (2023) also show that sample-wise monotonicity holds for isotropic linear regression given optimal dropout regularization. Kelly et al. (2022) study RF regression for time series, showing that more overparameterization strictly improves certain performance measures of interest in financial forecasting. It has also been argued that overparameterization provides benefits in terms of allowing efficient optimization (Jacot et al., 2018; Liu et al., 2022), network expressivity (Cybenko, 1989; Lu et al., 2017), and adversarial robustness (Bubeck & Sellke, 2021).

**The generalization of RF regression.** RF (ridge) regression was first proposed by Rahimi & Recht (2007) as a cheap approximation to KRR. Its generalization was first studied using classical capacity-based bounds Rahimi & Recht (2008); Rudi & Rosasco (2017). In the modern era, RF regression has seen renewed theoretical attention due to its analytical tractability and variable parameter count. Gerace et al. (2020) find closed-form equations for the test error of RF regression with a fixed projection matrix. Jacot et al. (2020b) show that the average RF predictor for a given dataset resembles a KRR predictor with greater ridge parameter. Mei & Montanari (2019); Mei et al. (2022) find closed-form equations for the average-case test error of RF regression in the special case of high-dimensional isotropic covariates. Maloney et al. (2022) find equations for the average test error of a general model of RF regression under a special "teacher equals student" condition on the task eigenstructure, and Bach (2023) similarly solved RF regression for the case of zero ridge. We report a general RF eigenframework that subsumes many of these closed-form solutions as special cases (see Appendix F). Our eigenframework can also be extracted, with some algebra, from replica calculations reported by Atanasov et al. (2023) (Section D.5.2) and Zavatone-Veth & Pehlevan (2023) (Proposition 3.1).

**Interpolation is optimal.** Many recent works have aimed to identify settings in which optimal generalization on noisy data may be achieved by interpolating methods, including local interpolating schemes (Belkin et al., 2018) and ridge regression (Liang & Rakhlin, 2020; Muthukumar et al., 2020; Koehler et al., 2021; Bartlett et al., 2020; Tsigler & Bartlett, 2023; Zhou et al., 2023). However, it is not usually the case in these works that (near-)interpolation is *required* to generalize optimally, as seen in practice. We argue that this is because these works focus entirely on the model, whereas one must also identify suitable conditions on the task being learned in order to make such a claim. Several papers have described ridge regression settings in which a *negative* ridge parameter is in fact optimal (Kobak et al., 2020; Wu & Xu, 2020; Tsigler & Bartlett, 2023). We consider only nonnegative ridge in this work to align with deep learning, but our findings are consistent with the task criterion found by Wu & Xu (2020).[2] In a similar spirit, Cheng et al. (2022) prove in a Bayesian linear regression setting that for low noise, algorithms must fit substantially below the noise floor to avoid being suboptimal.

## 3    PRELIMINARIES

We will work in a standard supervised setting: our dataset consists of $n$ samples $\mathcal{X} = \{x_i\}_{i=1}^n$ sampled i.i.d. from a measure $\mu_x$ over $\mathbb{R}^d$. We wish to learn a target function $f_*$ (which we assume to be square-integrable with respect to $\mu_x$), and are provided noisy training labels $\boldsymbol{y} = (y_i)_{i=1}^n$ where $y_i = f_*(x_i) + \mathcal{N}(0, \sigma^2)$ with noise level $\sigma^2 \geq 0$. Once a learning rule returns a predicted function $f$, we evaluate its train and test mean-squared error, given by $\mathrm{MSE}_{\mathrm{tr}} = \frac{1}{n} \sum_i (f(x_i) - y_i)^2$ and $\mathrm{MSE}_{\mathrm{te}} = \mathbb{E}_{x \sim \mu_x} \big[ (f(x) - f_*(x))^2 \big] + \sigma^2$ respectively.

---

[2]Some of our results — for example, Corollary 1 — give inequalities which, if satisfied, imply that zero ridge is optimal. It is generally the case that, when such an inequality is satisfied strictly (i.e. we do not have equality), a negative ridge would have been optimal had we allowed it.

RF regression is a learning rule defined by the following procedure. First, we sample $k$ weight vectors $\{w_i\}_{i=1}^{k}$ i.i.d. from some measure $\mu_w$ over $\mathbb{R}^p$. We then define the *featurization transformation* $\boldsymbol{\psi} : x \mapsto (g(w_i, x))_{i=1}^{k}$, where $g : \mathbb{R}^p \times \mathbb{R}^d \to \mathbb{R}$ is a feature function which is square-integrable with respect to $\mu_w$ and $\mu_x$. Finally, we perform standard linear ridge regression over the featurized data: that is, we output the function $f(x) = \boldsymbol{a}^T \boldsymbol{\psi}(x)$, where the weights $\boldsymbol{a}$ are given by $\boldsymbol{a} = (\boldsymbol{\Psi}\boldsymbol{\Psi}^T + \delta k \boldsymbol{I}_k)^{-1} \boldsymbol{\Psi} \boldsymbol{y}$ with $\boldsymbol{\Psi} = [\boldsymbol{\psi}(x_1), \cdots, \boldsymbol{\psi}(x_n)]$ and $\delta \geq 0$ a ridge parameter. If the feature function has the form $g(w, x) = h(w^T x)$ with $d = p$ and some nonlinearity $h$, then this model is precisely a shallow neural network with only the second layer trained.

RF regression is equivalent to kernel ridge regression

$$f(x) = \hat{\boldsymbol{k}}_{x\mathcal{X}}(\hat{\boldsymbol{K}}_{\mathcal{X}\mathcal{X}} + \delta \boldsymbol{I}_n)^{-1}\boldsymbol{y}, \tag{1}$$

where the vector $[\hat{\boldsymbol{k}}_{x\mathcal{X}}]_i = \hat{K}(x, x_i)$ and matrix $[\hat{\boldsymbol{K}}_{\mathcal{X}\mathcal{X}}]_{ij} = \hat{K}(x_i, x_j)$ contain evaluations of the (stochastic) random feature kernel $\hat{K}(x, x') = \frac{1}{k}\sum_i g(w_i, x)g(w_i, x')$. Note that as $k \to \infty$, the kernel converges in probability to its expectation $\hat{K}(x, x') \xrightarrow{k} K(x, x') := \mathbb{E}_w[g(w, x)g(w, x')]$ and RF regression converges to KRR with the deterministic kernel $K$.

### 3.1 SPECTRAL DECOMPOSITION OF $g$ AND THE GAUSSIAN UNIVERSALITY ANSATZ

Here we say what we mean by "task eigenstructure" in RF regression. Consider the bounded linear operator $T : L^2(\mu_w) \to L^2(\mu_x)$ defined as

$$(Tv)(x) = \int_{\mathbb{R}^p} v(w)g(w, x)d\mu_w(w).$$

The operator $T$ is a Hilbert-Schmidt operator to which the singular value decomposition can be applied to (Kato, 2013). That is, there is an orthonormal basis $(\zeta_i)_{i=1}^{\infty}$ of $(\mathrm{Ker}\, T)^{\perp} \subseteq L^2(\mu_w)$ and an orthonormal basis $(\phi_i)_{i=1}^{\infty}$ of $L^2(\mu_x)$ such that $T\zeta_i = \sqrt{\lambda_i}\phi_i$. Here $\{\lambda_i\}_{i=1}^{\infty}$ are the non-negative eigenvalues indexed in decreasing order and $\{\phi_i\}_{i=1}^{\infty}$ are the corresponding eigenfunctions of the integral operator $\Sigma : L^2(\mu_x) \to L^2(\mu_x)$ given by

$$(\Sigma u)(y) = \int_{\mathbb{R}^d} u(x)K(x, y)d\mu_x(x).$$

If we denote $T^{\star} : L^2(\mu_x) \to L^2(\mu_w)$ as the adjoint of $T$, then $\Sigma = TT^{\star}$. Moreover, the feature function $g$ admits the decomposition $g(w, x) = \sum_i \sqrt{\lambda_i}\zeta_i(w)\phi_i(x)$, where the convergence is in $L^2(\mu_w \otimes \mu_x)$. The decomposition of the deterministic kernel $K$ is given by

$$K(x, y) = \mathbb{E}_w[g(w, x)g(w, x')] = \sum_i \lambda_i \phi_i(x)\phi_i(y).$$

Note that the learning problem is specified entirely by $(n, k, \delta, \{\lambda_i\}, \{v_i\}, \{\zeta_i\}, \{\phi_i\})$.

The functions $\{\zeta_i\}$, $\{\phi_i\}$ viewed as random variables induced by $\mu_w$ and $\mu_x$ respectively can have a complicated distribution. However, since the functions form orthonormal bases, we know that the random variables are uncorrelated and have second moments equal to one — that is, $\mathbb{E}_{w \sim \mu_w}[\zeta_i(w)\zeta_j(w)] = \mathbb{E}_{x \sim \mu_x}[\phi_i(x)\phi_j(x)] = \delta_{ij}$. To make progress, throughout the text we will use the following *Gaussian universality ansatz* and assume that the distributions may be treated as uncorrelated Gaussians:

**Assumption A** (Gaussian universality ansatz). *The expected train and test error are unchanged if we replace $\{\zeta_i\}$, $\{\phi_i\}$ by random Gaussian functions $\{\tilde{\zeta}_i\}$, $\{\tilde{\phi}_i\}$ such that when sampling $w \sim \mu_w$ the values $\{\tilde{\zeta}_i(w)\}$ are i.i.d. samples from $\mathcal{N}(0, 1)$, and likewise for $x \sim \mu_x$ and $\{\tilde{\phi}_i(x)\}$.*

Assumption A seems strong at first glance. It is made plausible by many comparable universality results in random matrix theory which state that, when computing certain scalar quantities derived from large random matrices, only the first two moments of the elementwise distribution matter (up to some technical conditions), and the elements can thus be replaced by Gaussians for more convenient analysis (Davidson & Szarek, 2001; Tao, 2023). Assumption A holds provably for RF regression in certain asymptotic settings — see, for example, (Mei & Montanari, 2019; Mei et al., 2022). Most encouragingly, recent results for the test error of KRR derived using a comparable condition show excellent agreement with the test error computed on real data at moderate sample size (Sollich, 2001;

Bordelon et al., 2020; Jacot et al., 2020a; Simon et al., 2021; Loureiro et al., 2021; Wei et al., 2022), so we might expect to observe a similar universality in RF regression. Indeed, we will shortly validate Assumption A empirically, demonstrating excellent agreement between predictions for Gaussian statistics and RF experiments with real data (Figure 1). Given that our ultimate goal is to understand learning from real data, this empirical agreement is reassuring.

Under this universality ansatz, the statistics of the eigenfunctions may be neglected, and the learning task is specified entirely by the remaining variables $(n, k, \delta, \{\lambda_i\}, \{v_i\})$. We will now give a set of closed-form equations which estimate train and test error in terms of these quantities.

## 4 MORE DATA AND FEATURES ARE BETTER IN RF REGRESSION

### 4.1 THE OMNISCIENT RISK ESTIMATE FOR RF REGRESSION

In this section, we will first state our risk estimate for RF regression, which will enable us to conclude that more data and features are better.

---

Let $\kappa, \gamma \geq 0$ be the unique nonnegative scalars such that

$$n = \sum_i \frac{\lambda_i}{\lambda_i + \gamma} + \frac{\delta}{\kappa} \qquad \text{and} \qquad k = \sum_i \frac{\lambda_i}{\lambda_i + \gamma} + \frac{k\kappa}{\gamma}. \tag{2}$$

Let $z = \sum_i \frac{\lambda_i}{\lambda_i + \gamma}$ and $q = \sum_i \left( \frac{\lambda_i}{\lambda_i + \gamma} \right)^2$. The test and train error of RF regression are given approximately by

$$\mathrm{MSE}_{\mathrm{te}} \approx \mathcal{E}_{\mathrm{te}} = \frac{1}{1 - \frac{q(k-2z)+z^2}{n(k-q)}} \left[ \sum_i \left( \frac{\gamma}{\lambda_i + \gamma} - \frac{\kappa \lambda_i}{(\lambda_i + \gamma)^2} \frac{k}{k-q} \right) v_i^2 + \sigma^2 \right], \tag{3}$$

$$\mathrm{MSE}_{\mathrm{tr}} \approx \mathcal{E}_{\mathrm{tr}} = \frac{\delta^2}{n^2 \kappa^2} \mathcal{E}_{\mathrm{te}}. \tag{4}$$

---

Following Breiman & Freedman (1983); Wei et al. (2022), we refer to Equation 3 as the *omniscient risk estimate* for RF regression because it relies on ground-truth information about the task (i.e., the true eigenvalues $\{\lambda_i\}$ and target eigencoefficients $\{v_i\}$). We refer to the boxed equations, together with estimates for the bias, variance, and expectation of $f$ given in Appendix E.5, as the *RF eigenframework*. Several comments are now in order.

Like the framework of Bach (2023), our RF eigenframework is expected to become exact (with the error hidden by the "$\approx$" going to zero) in a proportional limit where $n$, $k$, and the number of eigenvalues diverge together. [3] However, we will soon see that it agrees well with experiment even at modest $n, k$. This eigenframework generalizes and unifies several known results which we elaborate upon in Appendix F.

**Deriving the omniscient risk estimate for RF regression.** We give a complete derivation of our RF eigenframework in Appendix E and give a brief sketch here. It is obtained from the fact that RF regression is KRR (c.f. Equation 1) with a stochastic kernel. As discussed in the introduction, many recent works have converged on an omniscient risk estimate for KRR under the universality ansatz, and if we knew certain eigenstatistics of the stochastic RF kernel $\hat{K}$, we could simply insert them into this known estimate for KRR and be done. Our main effort is in estimating these eigenstatistics using a handful of random matrix theory facts, after which we may read off the omniscient risk estimate for RF regression. Our derivation is nonrigorous, but we conjecture that it can be made rigorous with explicit error bounds decaying with $n, k$ as in Cheng & Montanari (2022).

**Plan of attack.** Our approach for the rest of the paper will be to rigorously prove facts about the deterministic quantities $\mathcal{E}_{\mathrm{te}}$ and $\mathcal{E}_{\mathrm{tr}}$ given by our omniscient risk estimates in Equations (3) and (4).

### 4.2 THE "MORE IS BETTER" PROPERTY OF RF REGRESSION

We now state the main result of this section.

---

[3]More precisely: in this limit, one can consider taking $n \to \eta n$, $k \to \eta k$, and duplicating each eigenmode $\eta$ times for an integer $\eta$, and then taking $\eta \to \infty$. Alternatively, new eigenmodes can be sampled from a fixed spectral distribution.

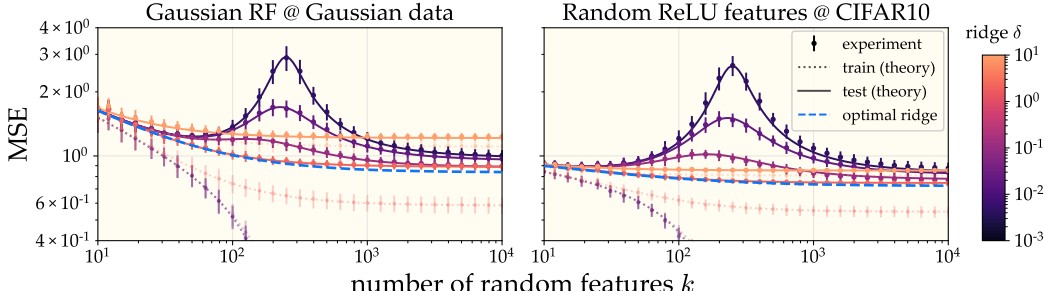

Figure 1: **At optimal ridge, more features monotonically improves test performance.** Train and test errors for RF regression closely match Equations (3) and (4) for both synthetic Gaussian data and CIFAR10 with random ReLU features. Plots show traces with $n = 256$ samples and varying number of features $k$. See Appendix A for experimental details and more plots.

---

**Theorem 1** (More is better for RF regression). *Let $\mathcal{E}_{\text{te}}(n, k, \delta)$ denote $\mathcal{E}_{\text{te}}$ with $n$ samples, $k$ features, and ridge $\delta$ with any task eigenstructure $\{\lambda_i\}_{i=1}^{\infty}, \{v_i\}_{i=1}^{\infty}$. Let $n' \geq n \geq 0$ and $k' \geq k \geq 0$. It holds that*

$$\min_{\delta'} \mathcal{E}_{\text{te}}(n', k', \delta') \leq \min_{\delta} \mathcal{E}_{\text{te}}(n, k, \delta), \tag{5}$$

*with strict inequality so long as $(n, k) \neq (n', k')$ and $\sum_i \lambda_i v_i^2 > 0$ (i.e., the target has nonzero learnable component).*

---

The proof, given in Appendix G, is elementary and follows directly from Equation 3. Theorem 1 states that the addition of either more data or more random features *can only improve generalization error* so long as we are free to choose the ridge parameter $\delta$. It is counterintuitive from the perspective of classical statistics, which warns against overparameterization: by contrast, we see that performance *increases* with additional overparameterization, with the limiting KRR predictor being the optimal learning rule. However, this is sensible if one views RF regression as a stochastic approximation to KRR: *the more features, the better the approximation to the ideal limiting process.* This interpretation lines up nicely with recent theoretical intuitions viewing infinite-width deep networks as noiseless limiting processes (Bahri et al., 2021; Atanasov et al., 2023; Yang et al., 2022).

### 4.3    EXPERIMENTS: VALIDATING THE RF EIGENFRAMEWORK

We perform RF regression experiments using real and synthetic data. Synthetic experiments use Gaussian data $x \sim \mathcal{N}(0, \text{diag}(\{\lambda_i\}))$ with $\lambda_i = i^{-1.5}$ and simple projection features $g(w, x) = w^T x$ with Gaussian weights $w \sim \mathcal{N}(0, \boldsymbol{I}_d)$. Experiments with real datasets use random ReLU features $g(w, x) = \text{ReLU}(w^T x)$ with Gaussian weights $w \sim \mathcal{N}(0, 2\boldsymbol{I}_d)$; the corresponding theoretical predictions use task eigenstructure extracted numerically from the full dataset. The results, shown in Figure 1 and elaborated in Appendix A, show excellent agreement between measured test and train errors and our theoretical predictions. The good match to real data justifies the Gaussian universality ansatz used to derive the framework (Assumption A).

## 5    OVERFITTING IS OBLIGATORY FOR KRR WITH POWERLAW EIGENSTRUCTURE

Having established that more data and more features are better for RF regression, we now seek an explanation for why "more fitting" — that is, little to no regularization — is also often optimal. As we now know that infinite-feature models are always best, in our quest for optimality we simply take $k \to \infty$ for the remainder of the paper and study the KRR limit. When $k \to \infty$, our RF eigenframework reduces to the well-established omniscient risk estimate for KRR, which we write explicitly in Appendix I.1. We will demonstrate that overfitting is obligatory for a class of tasks with powerlaw eigenstructure.

**Defining "overfitting."** How should one quantify the notion of "overfitting"? In some sense, we mean that the optimal ridge parameter $\delta_*$ which minimizes test error $\mathcal{E}_{\text{te}}$ is "small." However, $\delta_*$ will usually decay with $n$, so it is unclear how to define "small." In this work, we define overfitting via the *fitting ratio* $\mathcal{R}_{\text{tr/te}} := \mathcal{E}_{\text{tr}} / \mathcal{E}_{\text{te}} \in [0, 1]$, which has the advantage of remaining order unity even as $n$ diverges. The fitting ratio is a strictly increasing function of the ridge $\delta$, with $\mathcal{R}_{\text{tr/te}} = 0$ when $\delta = 0$ and $\mathcal{R}_{\text{tr/te}} \to 1$ as $\delta \to \infty$. Therefore, rather than minimizing $\mathcal{E}_{\text{te}}$ with respect to $\delta \in [0, \infty)$, we can

equivalently minimize $\mathcal{E}_{\text{te}}$ with respect to $\mathcal{R}_{\text{tr/te}} \in [0, 1)$. We will take the term "overfitting" to mean that $\mathcal{R}_{\text{tr/te}} \ll 1$.

**Definition 1** (Optimal ridge, test error, and fitting ratio). The optimal ridge $\delta_*$, optimal test error $\mathcal{E}_{\text{te}}^*$, and optimal fitting ratio $\mathcal{R}_{\text{tr/te}}^*$ are equal to the values of these quantities at the ridge that minimizes test error. That is,

$$\delta_* := \arg\min_\delta \mathcal{E}_{\text{te}}, \qquad \mathcal{E}_{\text{te}}^* = \mathcal{E}_{\text{te}}|_{\delta=\delta_*}, \qquad \mathcal{R}_{\text{tr/te}}^* = \mathcal{R}_{\text{tr/te}}|_{\delta=\delta_*}. \tag{6}$$

If the *optimal fitting ratio* $\mathcal{R}_{\text{tr/te}}^* := \arg\min_{\mathcal{R}_{\text{tr/te}}} \mathcal{E}_{\text{te}}$ is small — that is, if $\mathcal{R}_{\text{tr/te}}^* \ll 1$ — we may say that overfitting is optimal. If it is also true that any $\mathcal{R}_{\text{tr/te}} > \mathcal{R}_{\text{tr/te}}^*$ gives *higher* test error, we say that overfitting is *obligatory*.

In this section, we will describe a class of tasks with powerlaw eigenstructure for which overfitting is provably obligatory. For all powerlaw tasks, we will find that $\mathcal{R}_{\text{tr/te}}^*$ is bounded away from 1 (Theorem 2). Given an additional condition — namely, noise not too big, and an inequality satisfied by the exponents — we find that $\mathcal{R}_{\text{tr/te}}^* = 0$ (Corollary 1). Remarkably, when we examine real learning tasks with convolutional kernels, we will observe powerlaw structure in satisfaction of this obligatory overfitting condition, and indeed we will see that $\mathcal{R}_{\text{tr/te}}^* \approx 0$.

**Definition 2** ($\alpha, \beta$ powerlaw eigenstructure). A KRR task has $\alpha, \beta$ *powerlaw eigenstructure* for exponents $\alpha > 1, \beta \in (1, 2\alpha + 1)$ if there exists an integer $i_0 > 0$ such that, for all $i \geq i_0$, the task eigenvalues and eigencoeffients obey $\lambda_i = i^{-\alpha}$ and $v_i^2 = i^{-\beta}$.

In words, a task has $\alpha, \beta$ powerlaw eigenstructure if the task eigenvalues $\{\lambda_i\}_{i=1}^\infty$ and squared eigencoefficients $\{v_i^2\}_{i=1}^\infty$ have *powerlaw tails* with exponents $\alpha, \beta$. The technical condition $\beta < 2\alpha + 1$ needed for our proofs is mild, and we will find it is comfortably satisfied in practice.

**Target noise.** We will permit tasks to have nonzero noise. However, we must be mindful of a subtle but crucial point: *we do not want to take a fixed noise variance $\sigma^2 = \Theta(1)$, independent of $n$*. The reason is that the unlearned part of the signal will decay with $n$, so if $\sigma^2$ does *not* decay, the noise will eventually overwhelm the uncaptured signal. At this point, we might as well be training on pure noise. In this case, maximal regularization is optimal, and the model cannot possibly benefit from overfitting, as discussed in Section 1. We give a careful justification of this key point and compare with the benign overfitting literature in Appendix H.

We instead consider the setting where the noise $\sigma^2$ scales down in proportion to the uncaptured signal To do so, we set

$$\sigma^2 = \sigma_{\text{rel}}^2 \cdot \mathcal{E}_{\text{te}}|_{\sigma^2=\delta=0}, \tag{7}$$

where $\sigma_{\text{rel}}^2 = \Theta(1)$ is the *relative noise level* and $\mathcal{E}_{\text{te}}|_{\sigma^2=\delta=0}$ is the test error at zero noise and zero ridge. This scaling also has the happy benefit of simplifying many of our final expressions. (We note that this question of noise scaling will ultimately prove purely academic — when we turn to real datasets, we will find that all are very well described with $\sigma^2$ identically zero.)

We now state our main results.

---

**Theorem 2.** *Consider a KRR task with $\alpha, \beta$ powerlaw eigenstructure. Let the optimal fitting ratio and optimal test risk be given by Definition 1. At optimal ridge, the fitting ratio is*

$$\mathcal{R}_{\text{tr/te}}^* = r_*^2 + O(n^{-\gamma}) \tag{8}$$

*where $r_*$ is either the unique solution to*

$$\alpha - \beta - (\alpha - 1)\beta r_* + \alpha(\alpha - 1)(1 - r_*)^\beta \sigma_{\text{rel}}^2 = 0 \tag{9}$$

*over $r_* \in [0, 1)$ or else zero if no such solution exists, and $\gamma = \min(1, 2\alpha + 1 - \beta)$. Furthermore, this fitting ratio is the unique optimum (up to error decaying in $n$) in the sense that*

$$\frac{\mathcal{E}_{\text{te}}}{\mathcal{E}_{\text{te}}^*} \geq 1 + C_\alpha \left(\sqrt{\mathcal{R}_{\text{tr/te}}} - \sqrt{\mathcal{R}_{\text{tr/te}}^*}\right)^2 + O(n^{-\gamma}) \tag{10}$$

*where $C_\alpha = \frac{(\alpha-1)^2}{\alpha^2}$.*

---

The first part of Theorem 2 gives an equation whose solution is the optimal fitting ratio $\mathcal{R}^*_{\text{tr/te}}$ under powerlaw eigenstructure. The second part is a strong-convexity-style guarantee that, unless we are indeed tuned near $\mathcal{R}^*_{\text{tr/te}}$, we will obtain test error $\mathcal{E}_{\text{te}}$ worse than optimal by a constant factor greater than one. The proof of Theorem 2 consists of direct computation of $\mathcal{E}_{\text{te}}$ at large $n$, together with the observation that $\mathcal{R}_{\text{tr/te}} = \frac{\delta^2}{n^2 \kappa^2}$. The difficulty lies largely in the technical task of bounding error terms. We give the proof in Appendix I.

The optimal fitting ratio $\mathcal{R}^*_{\text{tr/te}}$ given by Theorem 2 will always be bounded away from 1. This implies that *some* overfitting will always be obligatory in order to reach near-optimal test error. The following corollary, which follows immediately from Theorem 2, gives a necessary and sufficient condition under which $\mathcal{R}^*_{\text{tr/te}} \approx 0$ and interpolation (i.e. zero ridge) is obligatory.

---

**Corollary 1.** *Consider a KRR task with $\alpha, \beta$ powerlaw eigenstructure. The optimal fitting ratio vanishes — that is, $\mathcal{R}^*_{\text{tr/te}} \xrightarrow{n} 0$ — if and only if*

$$\sigma^2_{\text{rel}} \leq \frac{\beta - \alpha}{\alpha(\alpha - 1)}. \tag{11}$$

---

Corollary 1 gives an elegant picture of what makes a task interpolation-friendly. First, we must have $\beta \geq \alpha$; otherwise, the RHS of Equation 11 is negative. Larger $\beta$ means that the error decays faster with $n$ (Cui et al., 2021), so $\beta \geq \alpha$ amounts to a requirement that the task is sufficiently easy.[4] Second, we must have sufficiently small noise relative to the uncaptured signal $\mathcal{E}_{\text{te}}|_{\sigma^2=\delta=0}$. More noise is permissible when the difference $\beta - \alpha$ is greater, which is sensible since noise serves to make a task harder. The fact that zero regularization can be the unique optimum even with nonzero label noise is surprising from the perspective of classical statistics, which cautions against fitting below the noise level.

With zero noise, Theorem 2 simplifies to the following corollary for the optimal fitting ratio:

---

**Corollary 2.** *Consider a KRR task with $\alpha, \beta$ powerlaw eigenstructure. When $\sigma^2 = 0$, the optimal fitting ratio at large $n$ converges to*

$$\mathcal{R}^*_{\text{tr/te}} \xrightarrow{n} \begin{cases} \frac{(\alpha - \beta)^2}{(\alpha - 1)^2 \beta^2} & \text{if } \beta < \alpha, \\ 0 & \text{if } \beta \geq \alpha. \end{cases} \tag{12}$$

---

Corollary 2 implies in particular that even if $\beta$ is slightly smaller than $\alpha$, we will still find that $\mathcal{R}^*_{\text{tr/te}} \approx 0$.

## 5.1 Experiments: overfitting is obligatory for MNIST, SVHN and CIFAR10 image datasets

We ultimately set out to understand an empirical phenomenon: the optimality of interpolation in many deep learning tasks. Having proposed a model for this phenomenon, is crucial that we now turn back to experiment and put it to the empirical test. We do so now with standard image datasets learned by convolutional neural tangent kernels. Running KRR with varying amounts of regularization, we observe that $\mathcal{R}^*_{\text{tr/te}} \approx 0$ and (near-)interpolation is indeed obligatory for all three datasets (Figure 2). This is due to both favorable task structure and low intrinsic noise: as we add artificial label noise, $\mathcal{R}^*_{\text{tr/te}}$ is no longer zero, in accordance with Corollary 1. We also find that adding label noise increases $\mathcal{R}^*_{\text{tr/te}}$ in accordance with our theory.

Even more remarkably, we find an excellent quantitative fit to our Lemma 10, which predicts $\mathcal{E}_{\text{te}}$ as a function of $\mathcal{R}_{\text{tr/te}}, \alpha, \beta, \sigma^2_{\text{rel}}$ up to a multiplicative constant. *This surprising agreement attests that, insofar as the effect of regularization is concerned, the relevant structure of these datasets can be summarized by just the two scalars $\alpha, \beta$.* These experiments resoundingly validate our theoretical picture for the examined tasks.

**Measuring the exponents $\alpha, \beta$.** We examine three standard image datasets — MNIST, SVHN, and CIFAR10 — and run KRR with the Myrtle convolutional neural tangent kernel (Shankar et al., 2020).

---

[4]More precisely, since a larger $\alpha$ corresponds to stronger inductive biases, $\beta \geq \alpha$ means that in some sense "the task is no harder than the model anticipated."

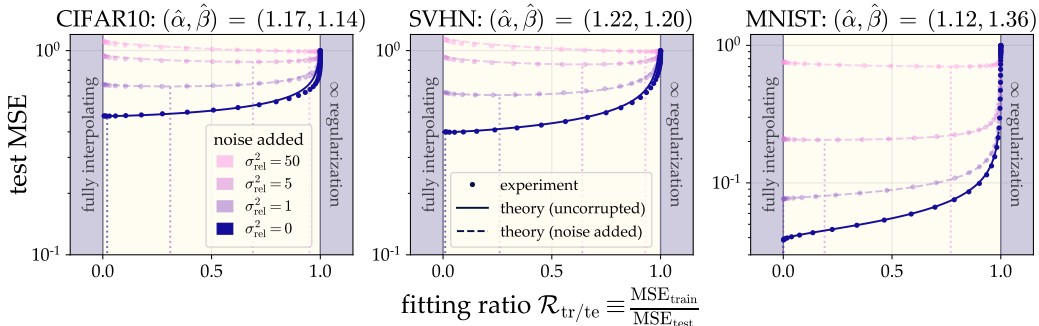

Figure 2: **Overfitting is obligatory in standard computer vision tasks.** We run KRR with convolutional NTKs on three tasks using varying ridge parameter and label noise, measuring test error and the fitting ratio $\mathcal{R}_{\mathrm{tr/te}}$. We then compare to theoretical predictions (c.f. Lemma 10) computed from measured powerlaw exponents $\hat{\alpha}$, $\hat{\beta}$. When no noise is added, we observe that the optimal fitting ratio is $\mathcal{R}^*_{\mathrm{tr/te}} \approx 0$ (blue vertical dotted line) and (near-)interpolation is required to achieve optimal error. These tasks have low intrinsic noise, and as label noise is added, $\mathcal{R}^*_{\mathrm{tr/te}}$ becomes nonzero, as predicted by Corollary 1. Curves with noise added are rescaled to preserve total task power. See Appendix B for exponent measurements and Appendix D for full experimental details.

We also report results for F-MNIST in Appendix D. We wish to check for powerlaw decay $\lambda_i \propto i^{-\alpha}$ and $v_i^2 \propto i^{-\beta}$ and estimate the exponents $\alpha, \beta$.

We measure $\alpha$ using the method of Wei et al. (2022). Assuming $\lambda_i \propto i^{-\alpha}$, it is easily shown that at zero ridge, the implicit regularization constant $\kappa$ in the eigenframework decays with $n$ as $\kappa(n) \asymp n^{-\alpha}$ (see e.g. our Lemma 2). Wei et al. (2022) show, theoretically under Gaussian universality and empirically for real data, that $\kappa(n)$ is well approximated by $\kappa(n) \approx \mathrm{Tr}[\boldsymbol{K}_n^{-1}]^{-1}$ where $\boldsymbol{K}_n$ is the empirical kernel matrix on $n$ samples. An estimate $\hat{\alpha}$ of the true exponent $\alpha$ may thus be extracted by plotting many points $(n, \mathrm{Tr}[\boldsymbol{K}_n^{-1}]^{-1}) \in \mathbb{R}^2$ on a log-log plot and fitting a line.

To measure $\beta$, we make use of the eigenframework prediction that, with $n$ samples, zero ridge, and zero noise, the test risk decays as $\mathtt{MSE}_{\mathrm{te}}(n) \approx \mathcal{E}_{\mathrm{te}}(n) \asymp n^{-(\beta-1)}$ (see e.g. Cui et al. (2021) or our Lemma 3). Like with $\alpha$, we can thus extract an estimate $\hat{\beta}$ of the true exponent $\beta$ by plotting many points $(n, \mathtt{MSE}_{\mathrm{te}}(n)) \in \mathbb{R}^2$ on a log-log plot and fitting a line.

## 6 DISCUSSION

The present work is part of the research program aiming to understand the shortcomings of classical learning theory and to develop analyses suitable to machine learning as it exists today. We have presented two tractable models capturing the "more is better" spirit of deep learning, but we cannot consider this quest done until we have not only transparent models but also coherent new intuitions to take the place of appealing but outdated classical ones. To that end, we propose a few here.

One once-well-believed classical nugget of wisdom is the following: *Overparameterization is harmful because it permits models to express complicated, poorly-generalizing solutions*. Taking inspiration from our RF models, perhaps we ought instead to believe the following: ***Overparameterization is desirable because bigger models more closely approximate an ideal limiting process with infinitely many parameters.*** Overparameterization *does* permit a model to *express* complicated, poorly-generalizing solutions, but it *does not* mean that it will *actually learn* such a thing: models are simple creatures and tend not to learn unnecessarily complication without a good reason to.

Another classical view is that *interpolation is undesirable because if a model interpolates, it has fit the noise, and so will generalize poorly.* Our story with KRR suggests that perhaps we should instead hold the following belief: ***So long as the inductive bias of the model is a good match for the task and the noise is not too large, additional regularization is unnecessary, and it is optimal to fit with train error much less than test error.***

ACKNOWLEDGEMENTS

JS thanks Alex Maloney, Alex Wei, Ben Adlam, Berfin Şimşek, Blake Bordelon, Jacob Zavatone-Veth, Francis Bach, Michael DeWeese, and Theo Misiakiewicz for useful discussions on the generalization of RF regression, as well as Roman Novak and Song Mei for useful discussions on the eigenstructure of image tasks early in the development of our empirical measurements. JS used Zymbolic for rapid math typesetting. JS gratefully acknowledges support from the National Science Foundation Graduate Fellow Research Program (NSF-GRFP) under grant DGE 1752814. MB acknowledges support from National Science Foundation (NSF) and the Simons Foundation for the Collaboration on the Theoretical Foundations of Deep Learning (https://deepfoundations.ai/) through awards DMS-2031883 and #814639 and the TILOS institute (NSF CCF-2112665). This work used the programs (1) XSEDE (Extreme science and engineering discovery environment) which is supported by NSF grant numbers ACI-1548562, and (2) ACCESS (Advanced cyberinfrastructure coordination ecosystem: services & support) which is supported by NSF grants numbers #2138259, #2138286, #2138307, #2137603, and #2138296. Specifically, we used the resources from SDSC Expanse GPU compute nodes, and NCSA Delta system, via allocations TG-CIS220009.

SUMMARY OF CONTRIBUTIONS

JS developed the main theoretical results, wrote the manuscript, and led the team logistically. DK developed and performed all experiments and assisted in writing. NG aided in framing the overall narrative, assisted in writing, and provided key technical insights during the development of the theory. MB proposed this line of study and provided overarching guidance throughout the project.

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

# A EXPERIMENTAL VALIDATION OF OUR EIGENFRAMEWORK FOR RF REGRESSION

We validate our RF eigenframework by comparing its predictions against two examples of random feature model:

**Random Gaussian projections.** We draw latent data vectors $x \sim \mathcal{N}(0, I_m)$ from an isotropic Gaussian in a high dimensional ambient space (dimension $m = 10^4$).[5] The target function is linear as $y = v^T x + \xi$, where the target coefficients $v_i$ follow a powerlaw $v_i = \sqrt{i^{-\beta}}$ with $\beta = 1.5$, and $\xi \sim \mathcal{N}(0, \sigma^2)$ is Gaussian label noise with $\sigma^2 = 0.5$.

We construct random features as $\psi(x) = W\Lambda^{1/2}x$. Here, $W \in \mathbb{R}^{k \times m}$ has elements drawn i.i.d. from $\mathcal{N}(0, \frac{1}{k})$. $\Lambda = \mathrm{diag}(\lambda_1 \ldots \lambda_m)$ is a diagonal matrix of kernel eigenvalues, $\lambda_i = i^{-\alpha}$ with $\alpha = 1.5$. With this construction, the full-featured kernel matrix is (in expectation over the features) $\mathbb{E}_W[K] = X^T\Lambda X$, with $[X] = [x_1, \cdots, x_n]$.

**Random ReLU features.** CIFAR10 input images are normalized to global mean 0 and standard deviation 1. The labels are binarized (with $y = \pm 1$) into two classes: *things one can ride* (airplane, automobile, horse, ship, truck) and *things one ought not to ride* (bird, cat, deer, dog, frog). Thus the target function is scalar.

The features are given by $\psi(x) = \mathrm{ReLU}(W^T x)$ where $W \in \mathbb{R}^{k \times m}$ has elements drawn i.i.d. from $\mathcal{N}(0, \frac{2}{d_{\mathrm{in}}})$, with $d_{\mathrm{in}} = 3072$. With this construction, the limiting infinite-feature kernel is in fact the "NNGP kernel" of an infinite-width 1-hidden-layer ReLU network (Neal, 1996; Lee et al., 2018).

**Theoretical predictions.** The RF framework is an omniscient risk estimate, so to use it, we must have on hand the eigenvalues of the infinite-feature kernel $K$ and the eigencoefficients of the target function w.r.t to $K$. For the synthetic data, we dictate the eigenstructure by construction: $\lambda_i = i^{-1.5}$ and $v_i^2 = i^{-1.5}$. For random ReLU features, we use the `neural tangents` library (Novak et al. (2020)) to compute the NNGP kernel matrix of CIFAR10, and then diagonalize it to extract the eigenstructure. (We diagonalize an $n = 30000$ subset of the kernel matrix since this is the largest matrix we can diagonalize on a single A100 GPU without resorting to distributed eigensolvers.)

When evaluating our eigenframework, we numerically solve Equation 2 for $\kappa$ and $\gamma$. This can prove a slightly finicky process. We use an inner-loop outer-loop routine as follows. In the inner loop, $\kappa$ is fixed, we solve for $\gamma$ such that $n = \sum_i \frac{\lambda_i}{\lambda_i + \gamma} + \frac{\delta}{\kappa}$, and we return the error signal $k - \sum_i \frac{\lambda_i}{\lambda_i + \gamma} - \frac{k\kappa}{\gamma}$, equal to the discrepancy in the other equation. In the outer loop, we optimize $\kappa$ to drive that error signal to zero.

**Experimental details.** We vary $n \in [10^1, 10^4]$, $k \in [10^1, 10^4]$, $\delta \in [10^{-3}, 10^2]$. We perform 45 trials of each experimental run at a given $(n, k, \delta)$; however, in each trial we fix the size-$n$ dataset as we vary the random features.

**Additional plots.** We report additional comparisons between RF experiments and our theory in Figures 3 and 4.

**Code availability.** Code to reproduce all experiments is available at `https://github.com/dkarkada/more-is-better`.

---

[5]In the main text, we purported to draw the data anisotropically as $x \sim \mathcal{N}(0, \Lambda)$. For our explanation here, we introduce $\Lambda$ at the random projection stage, which amounts to the same thing.

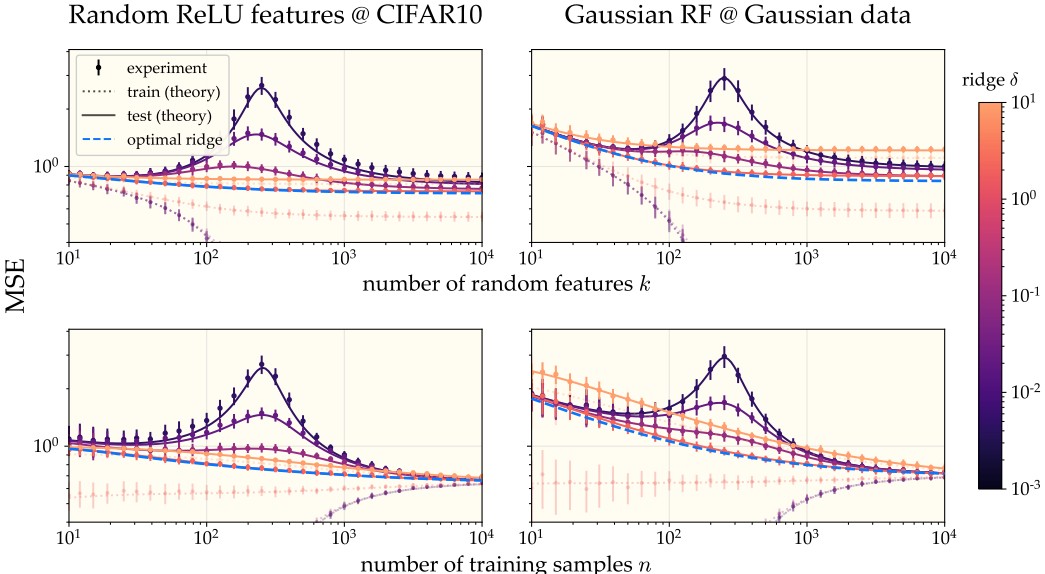

Figure 3: **Empirical verification of the RF eigenframework.** We plot various traces of train and test error, both experimental and theoretical as predicted by Equations (3) and (4), for two random feature models. (**top row, same as Figure 1**) We fix the trainset size $n = 256$ and vary the number of features $k$. (**bottom row**) We fix the number of random features $k = 256$ and vary the training set size $n$. Note that in this row, the classical underparametrized regime is to the *right* of the interpolation threshold, and the modern overparametrized regime is to the left.

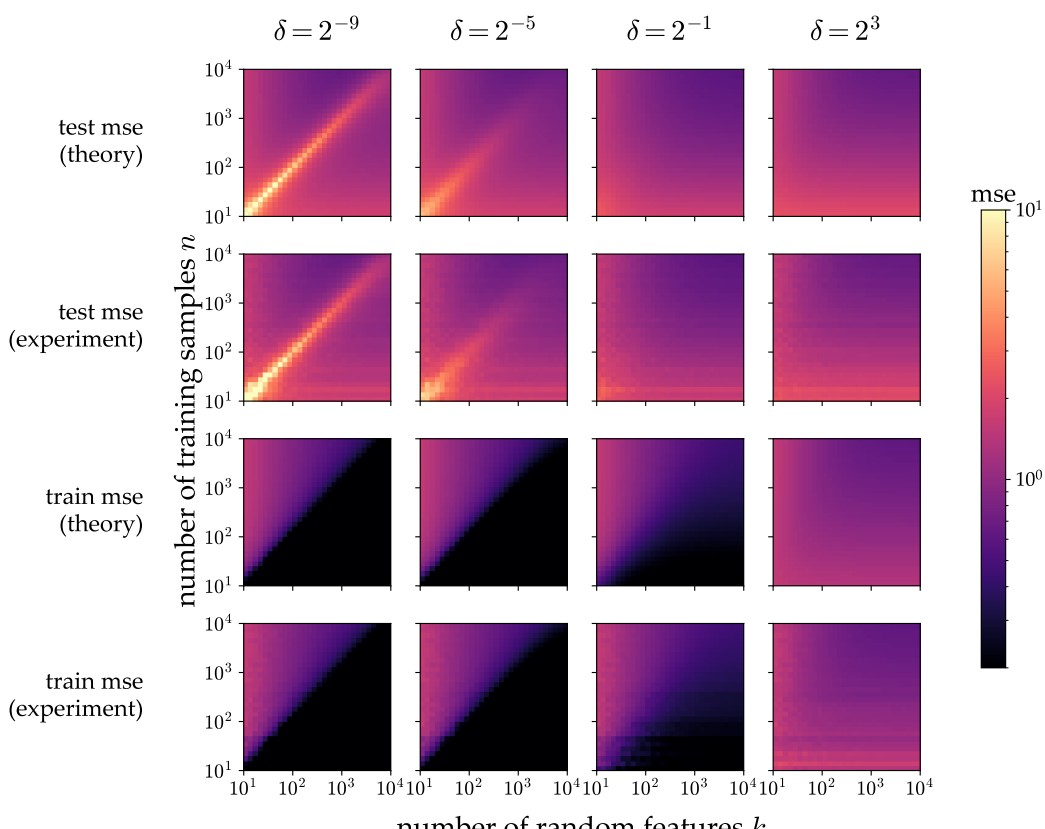

Figure 4: For RF regression with synthetic data, we show heatmaps of average train and test MSE as a function of training set size $n$ and number of random features $k$. We vary the ridge parameter $\delta$ from underregularized (left column) to overregularized (right column). In the underregularized setting, the signature double descent peak (bright diagonal) separates the classical regime (upper triangle) from the modern interpolating regime (lower triangle). In the overregularized setting, the model fails to interpolate the training data even at low $n$. Our theory accurately captures these phenomena. Note: at each $n$, we use the same batch of random datasets for all $k$, resulting in horizontal stripes visible at low $n$ that may be ignored as artifacts.

## B  TECHNIQUES FOR MEASURING POWERLAW EXPONENTS IN KRR TASKS

Our analysis of overfitting in kernel regression relies on an assumption of "powerlaw eigenstructure" (Definition 2) characterized by two exponents $\alpha, \beta$. Here we describe our procedures for measuring $\alpha, \beta$ for real datasets.

**Extracting $\alpha$ from the effective regularization $\kappa(n)$.**

The KRR eigenframework on which our results are based entails the computation of an intermediate quantity $\kappa$ which serves as an "effective regularization constant" (Simon et al., 2021; Bordelon et al., 2020; Jacot et al., 2020a; Wei et al., 2022). This quantity decreases as more data are added: writing $\kappa$ as a function of $n$, one usually finds decay as $\kappa(n) \asymp \lambda_n$.[6] In our case, if indeed we have $\lambda_i \propto i^{-\alpha}$ for large $i$, then we expect that $\kappa(n) \asymp i^{-\alpha}$.

Fortunately, Wei et al. (2022) describe a method by which $\kappa(n)$ may be experimentally measured. Under a reasonable Gaussian universality assumption, they find that $\kappa(n) \approx \mathrm{Tr}[\boldsymbol{K}_n^{-1}]^{-1}$, where $\boldsymbol{K}_n$ is an empirical kernel matrix computed from $n$ samples. When we plot points $(n, \mathrm{Tr}[\boldsymbol{K}_n^{-1}]^{-1})$ on a log-log plot for many values of $n$ for the four datasets we study, we indeed see a clear linear tail indicative of powerlaw decay of $\kappa(n)$ and whose slope $\hat{\alpha}$ we can easily extract after performing a linear fit. We show these linear fits in Figure 5.

**Extracting $\beta$ from the test error $\mathtt{MSE}_{\mathrm{te}}(n)$.**

As discussed in the main text, we generally expect true risk at zero ridge and zero noise to decay proportionally to the uncaptured signal as $\mathtt{MSE}_{\mathrm{te}}(n) \approx \mathcal{E}_{\mathrm{te}}(n) \propto \sum_{i>n} v_i^2 \asymp n^{-(\beta-1)}$. For the four datasets we study, we indeed see a linear decay when we plot points $(n, \mathtt{MSE}_{\mathrm{te}}(n))$ on a log-log plot. We fit a line and extract $\hat{\beta}$ as the slope. We show these plots and linear fits in Figure 5.

**Intuitions about powerlaw eigenstructure.** Having measured powerlaw structure in several image datasets, we now give some informal discussion of how this structure might be interpreted. Informally, powerlaw eigenstructure describes the structure of natural data in two ways:

- The eigenvalues of the kernel roughly decay as a powerlaw: $\lambda_i \sim i^{-\alpha}$. (Equivalently, the spectrum of the covariance matrix of the data distribution *in the kernel's feature space* decay as a powerlaw.) Since $\lambda_i$ represents a kernel's "willingness" to learn eigenmode $i$, we may interpret $\alpha$ as representing the kernel's parsimony: in modeling the data, a kernel with large $\alpha$ tends to overattribute explanatory power to its top $n$ eigenmodes, confidently neglecting its tail eigenmodes.

- The target function, expressed as a vector in the kernel's eigenbasis, has components whose squares roughly decay as a powerlaw: $v_i^2 \sim i^{-\beta}$. Since larger $\beta$ implies a greater proportion of total task power in the top eigenmodes, we may interpret $\beta$ as representing the target function's comprehensibility (to the kernel learner): targets with large $\beta$ are easier to learn.

These informal interpretations suggest that a kernel is well-suited to learn a task if $\alpha$ is sufficiently small compared to $\beta$. Otherwise, the kernel is simply too parochial to learn the intricacies of the target function; such a kernel will generalize poorly in the absence of regularization. This intuition is made precise in Corollary 1.

*Remark.* Powerlaw eigenstructure is a remarkable constraint. There is no clear reason why arbitrary data distributions and target functions should have this structure, and yet we observe that natural data do. This fact is both a miracle and a blessing for theorists, as it strongly restricts the class of data distributions we need concern ourselves with to explain the behavior of deep neural networks. It remains a major open question to fully explain and characterize the powerlaw structure of natural data with respect to neural kernels.

---

[6]Mallinar et al. (2022) find that $\kappa(n)$ can in fact be off of $\lambda_n$ by log factors for certain exotic "benign overfitting" spectra, but this is beyond our scope here.

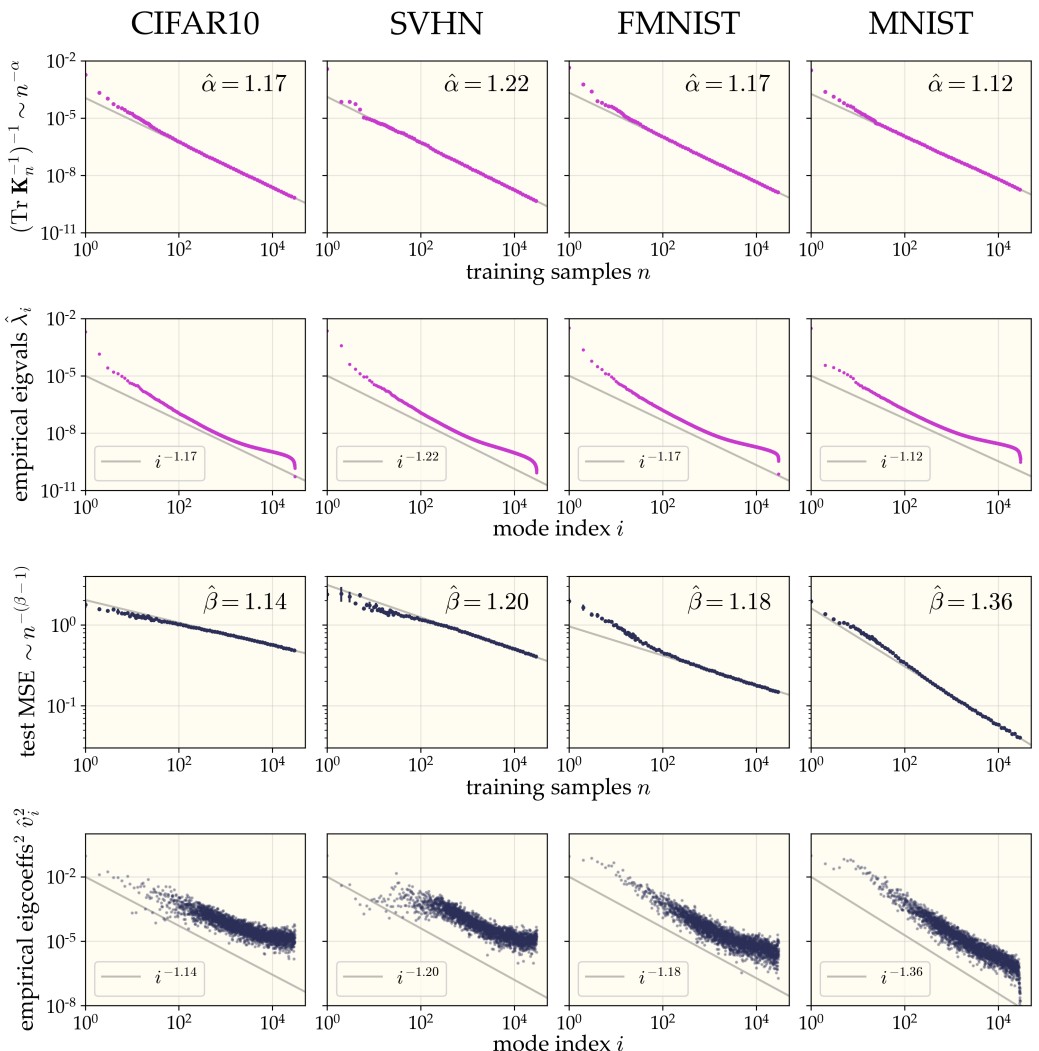

Figure 5: For the four vision tasks studied (columns), we show our techniques for measuring $\alpha$ (first row) and $\beta$ (third row). We fit the powerlaw decay in the tails (solid line) and report the corresponding exponent measurements (text). These plots are generated by studying increasingly large $n \times n$ training-data kernel matrices. For visual comparison, we include the empirical eigenstructure (eigenvalues and squared eigencoefficients of the *full* training-data kernel matrix, second and fourth rows respectively), along with a powerlaw decay with our measured exponent (solid line). Note that linear fits to the empirical eigenstructure (rows two and four) would be worse than to our proxy measurements (rows one and three).

## C COMPARISON: PROXY MEASUREMENTS OF $\alpha, \beta$ RECOVER TRUE EXPONENTS MORE ACCURATELY THAN MORE DIRECT MEASUREMENTS

Our theory for overfitting relies on the ansatz of powerlaw task eigenstructure that, at large $i$, task eigenvalues and eigencoefficients decay as $\lambda_i \propto i^{-\alpha}$ and $v_i^2 \propto i^{-\beta}$. Prior works, including Spigler et al. (2020); Lee et al. (2020); Bahri et al. (2021), extracted the exponents $\alpha, \beta$ from direct measurements of $\lambda_i$ and $v_i^2$ for full datasets. However, as detailed in Appendix B, we instead extract $\alpha, \beta$ more indirectly from the proxy quantities $\kappa$ and $\mathrm{MSE_{te}}$. The purpose of this appendix is to explain a flaw in the direct method and thereby motivate this proxy method.

We begin by describing the direct method. First, one computes an empirical kernel matrix $\boldsymbol{K}_n$ on a dataset of size $n$. Since $\boldsymbol{K}_n$ is real symmetric, we may numerically diagonalize it as $\boldsymbol{K}_n = \boldsymbol{\Phi}\hat{\boldsymbol{\Lambda}}\boldsymbol{\Phi}^T$, where $\boldsymbol{\Phi}\boldsymbol{\Phi}^T = n\boldsymbol{I}_n$ and $\hat{\boldsymbol{\Lambda}} = \mathrm{diag}(\hat{\lambda}_1, ..., \hat{\lambda}_n)$ is a diagonal matrix of $n$ estimated eigenvalues. One then projects the labels onto the kernel eigenvectors to find the target eigencoefficient vector $\hat{\boldsymbol{v}} = n^{-1}\boldsymbol{\Phi}^T\boldsymbol{y}$. Finally, one plots the computed eigenvalues and target eigencoefficients (or a tailsum thereof) on a log-log plot and performs linear fits, extracting powerlaw exponents from the slopes.

The essential problem with the direct method is *finite-sample-size effects* resulting from the finiteness of $n$. For example, the extracted eigenvalues $(\hat{\lambda}_i)_{i=1}^n$ may be viewed as an estimation of the first $n$ ground-truth eigenvalues $(\lambda_i)_{i=1}^n$, but one generally expects this estimation to be accurate only for indices $i \ll n$. In practice, we find it is often unclear for what range of indices we may say that "$i \ll n$" and that we may therefore trust our estimated eigenvalues. This is not a trivial problem: we find that the error and ambiguity thereby induced can be quite large!

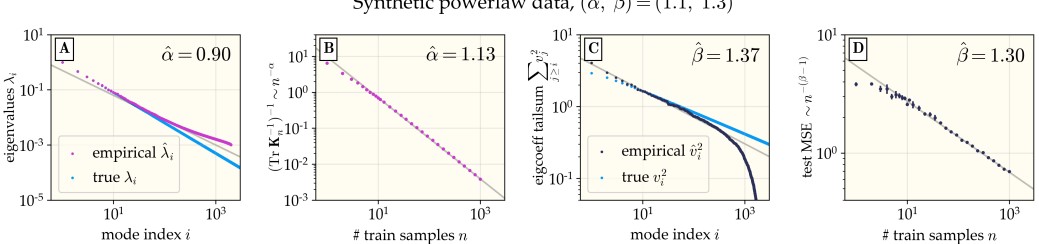

Synthetic powerlaw data, $(\alpha, \beta) = (1.1, 1.3)$

Figure 6: **Proxy methods for estimating $\alpha, \beta$ have lower error than direct powerlaw fits to task eigenstructure.** We generate $n = 2000$ samples of synthetic Gaussian data with ground-truth powerlaw eigenstructure $\lambda_i = i^{-1.1}$, $v_i^2 = i^{-1.3}$ and compare direct and proxy methods of estimating these exponents. **(A,C)** We diagonalize the empirical kernel, extracting estimated eigenvalues $(\hat{\lambda})_{i=1}^n$ and eigencoefficients $(\hat{v}_i)_{i=1}^n$. We plot the empirical eigenvalues and the tailsum of the empirical eigencoefficients. We observe that these do not clearly lie on a line on a log-log plot — a red flag — but we nonetheless depict reasonable attempts (gray lines) to fit lines. (For the eigenvalues, we do not use the first $\sim 20$ eigenvalues in our linear fit, since in practice the top eigenvalues usually do not obey the powerlaw in the tail and must be discarded.) The estimated exponents $(\hat{\alpha}, \hat{\beta}) = (0.90, 1.37)$ thus obtained have fairly high error from the ground-truth values. For visual comparison, we underlay the ground-truth eigenvalues and eigencoefficient tailsums (blue dots in both plots). **(B,D)** We estimate $\alpha, \beta$ from proxy measurements as described in Appendix B. The resulting estimates $(\hat{\alpha}, \hat{\beta}) = (1.13, 1.30)$ are much closer to the ground-truth values. The error in $\alpha$ is primarily due to the technical detail that our synthetic powerlaw data has a finite maximum index $i_{\max} = 3 \times 10^6$ for computational feasibility, whereas the theory on which this estimate is based assumes $i_{\max} = \infty$.

This is best illustrated with an example. To compare the accuracy of direct and proxy methods for exponent measurement, we construct a synthetic task with Gaussian data with powerlaw eigenstructure with $\alpha = 1.1$ and $\beta = 1.3$ and try out both methods for recovering these exponents. The results are illustrated in Figure 6. For eigenvalues, we find that *only the first few eigenvalues are recovered accurately,* and attempted fits to the middle or tail of the spectrum yield large measurement errors. This is important because, in practice, the first few eigenvalues typically do not follow the powerlaw of the tail, leaving the experimenter with few or no clean power eigenvalues to which to fit a line. However, our proxy measurement recovers a decent approximation to the true $\alpha$. For eigencoefficients, since individual eigencofficients generally look noisy and require smoothing to see a powerlaw, we

follow Spigler et al. (2020) and plot the tailsum $\sum_{j \geq i} v_j^2$ as a function of $i$, which is expected to decay as $i^{-(\beta-1)}$, the same exponent as that of test error. Here, too, we observe significant finite-size effects and are unable to observe the ground-truth powerlaw, but we recover the true $\beta$ with excellent accuracy from a plot of error vs. sample size.

## D    EXPERIMENTAL VALIDATION OF THEORY FOR POWERLAW TASKS

We compute neural tangent kernel matrices for the Myrtle-10 convolutional architecture on four standard computer vision datasets: CIFAR-10, Street View House Numbers, FashionMNIST, and MNIST. For each, we perform kernel regression varying the ridge and added label noise. We additionally measure the eigenstructure exponents $\alpha$ and $\beta$ using the techniques described in Appendix B. We use these measurements to predict the train and test error of these kernel learners and find excellent match with experiment (Figure 7).

**Experimental details.** We use the `neural tangents` library (Novak et al. (2020)) to compute the convolutional NTK matrices (CNTKs). It takes about four A100 GPU-days to compute each CNTK. We normalize each dataset to have global mean 0 and standard deviation 1. We do not binarize the labels: the learning task is one-hot 10-class regression. For experiments with label noise, the added noise is a Gaussian vector whose norm has *total* variance $\sigma^2 = \sigma_{\text{rel}}^2 \cdot \mathcal{E}_{\text{te}}|_{\sigma^2 = \delta = 0}$ (see Equation 7). After adding noise, we normalize all label vectors to have unit norm so that all curves in Figure 2 intersect at $(\mathcal{R}_{\text{tr/te}} = 1, \mathcal{E}_{\text{te}} = 1)$.

In Figure 2 and Figure 7, each theory curve is vertically shifted by a constant multiplicative prefactor which is chosen such that the theory curve agrees with the experimental data at $\mathcal{R}_{\text{tr/te}} = 0$. This post-hoc fit is required because the derivation of Lemma 10 assumes the eigencoefficients follow a powerlaw with prefactor unity (i.e., $v_i^2 = Ai^{-\beta}$ with $A = 1$), while the true eigencoefficients will be scaled, i.e. $v_i^2 = Ai^{-\beta}$ for some $A \neq 1$ which may not be easy to measure. For this reason, we choose to simply fit the prefactor. When looking at Figure 2, then, we get theory-experiment match at $\mathcal{R}_{\text{tr/te}} = 0$ for free, and the interesting fact is that we also have agreement for $\mathcal{R}_{\text{tr/te}} \in (0, 1)$.

**Code availability.** Code to reproduce all experiments is available at `https://github.com/dkarkada/more-is-better`.

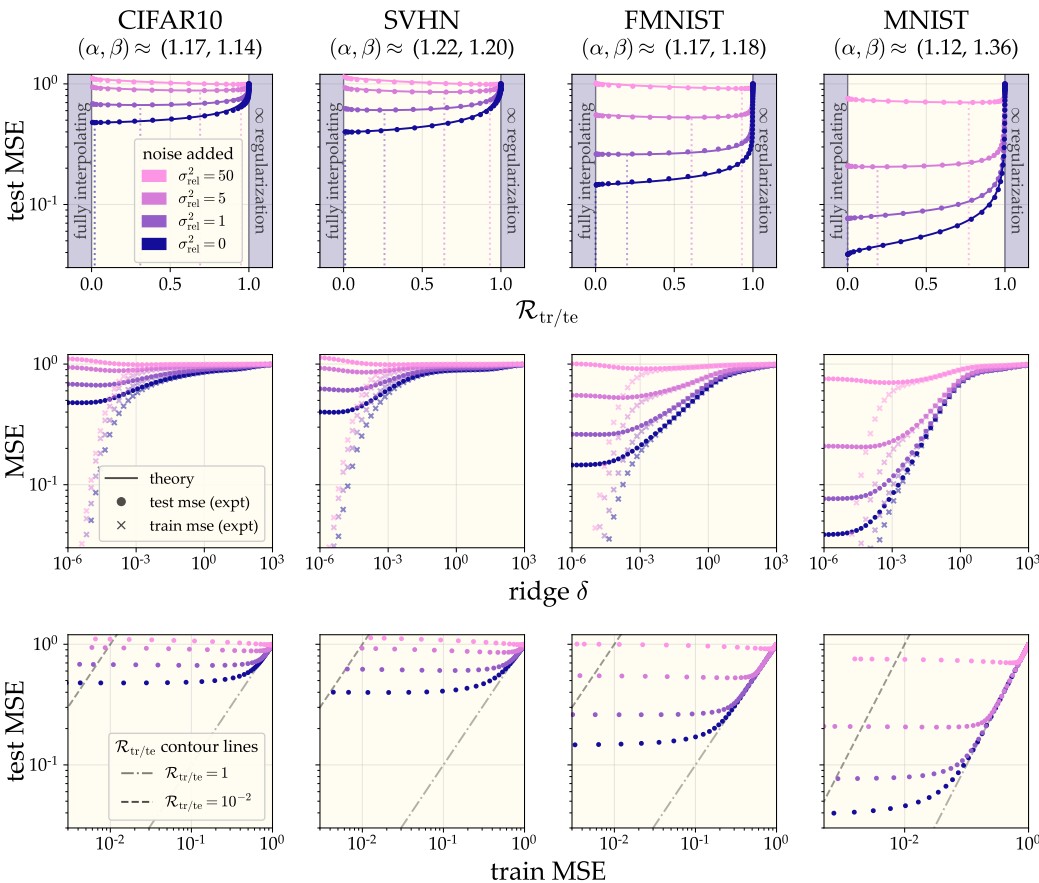

Figure 7: **Top row.** We extend the experiment shown in Figure 2 to the FashionMNIST dataset. Theoretical predictions using measured exponents accurately capture the obligatory overfitting phenomenon. **Middle row.** We plot experimental test and train error as a function of ridge. **Bottom row.** We plot experimental test error as a function of train error. Contours of constant $\mathcal{R}_{\mathrm{tr/te}}$ are parallel diagonals.

# E  DERIVATION OF THE RF EIGENFRAMEWORK

In this appendix, we give a derivation of the eigenframework giving the train and test risk of RF regression which we report in Section 4. Our plan of attack is as follows. First we will recall from the literature the comparable eigenframework for KRR, expressing it in a manner convenient for our task. We will then explicitly write RF regression as an instance of KRR with a stochastic kernel. This framing will make it clear that, if we could understand certain statistics of the (stochastic) eigenstructure of the RF regression kernel, we could directly plug them into the KRR eigenframework. We will then use a single asymptotic random matrix theory identity to compute the various desired statistics. Inserting them into the KRR eigenframework, we will arrive at our RF eigenframework.

Our derivation will be nonrigorous in that we will gloss over technical conditions for the applicability of the KRR eigenframework for the sake of simplicity. Nonetheless, we will have strong evidence that our final answer is correct by its recovery of known results in various limits of interest (Appendix F) and by its strong agreement with real data (Figure 1).

## E.1  RECALLING THE KRR EIGENFRAMEWORK.

We now state the omniscient risk estimate for KRR (or equivalently linear ridge regression) under Gaussian design which has been converged upon by many authors in recent years (Sollich, 2001; Bordelon et al., 2020; Jacot et al., 2020a; Simon et al., 2021; Loureiro et al., 2021; Dobriban & Wager, 2018; Wu & Xu, 2020; Hastie et al., 2022; Richards et al., 2021). We phrase the framework in a slightly different way than in Appendix I.1 which will be more suitable to our current agenda.

As in the main text, let the Mercer decomposition of the kernel $K$ be $K(x, x') = \sum_{i=1}^{\infty} \lambda_i \phi_i(x) \phi_i(x')$, where $\{\phi_i\}_{i=1}^{\infty}$ are a complete basis of eigenfunctions which are orthonormal with respect to the data measure $\mu_x$. We still assume our Gaussian universality ansatz (Assumption A) over the eigenfunctions $\{\phi_i\}_{i=1}^{\infty}$.

We will find it useful to pack the eigenvalues into the (infinite) matrix $\mathbf{\Lambda} = \mathrm{diag}(\lambda_1, \lambda_2, \ldots)$, the target eigencoefficients into the (infinite) vector $\mathbf{v} = (v_1, v_2, \ldots)$, and the set of eigenfunctions evaluated on any given data point into the (infinite) vector $\boldsymbol{\phi}(x) = (\phi_1(x), \phi_2(x), \ldots)$. Using this notation, the kernel function is given by

$$K(x, x') = \boldsymbol{\phi}(x)^T \mathbf{\Lambda} \boldsymbol{\phi}(x'). \tag{13}$$

The KRR eigenframework appearing in these prior works is as follows.[7] First, let $\kappa \geq 0$ be the unique nonnegative solution to the equation[8]

$$n = \mathrm{Tr}\left[\frac{\mathbf{\Lambda}}{\mathbf{\Lambda} + \kappa \mathbf{I}}\right] + \frac{\delta}{\kappa}. \tag{14}$$

Then test and train MSE are well-approximated by

$$\mathcal{E}_{\mathrm{te}} = \frac{n}{n - \mathrm{Tr}\left[\left(\frac{\mathbf{\Lambda}}{\mathbf{\Lambda} + \kappa \mathbf{I}}\right)^2\right]} \left(\mathbf{v}^T \left(\frac{\kappa}{\mathbf{\Lambda} + \kappa \mathbf{I}}\right)^2 \mathbf{v} + \sigma^2\right), \tag{15}$$

$$\mathcal{E}_{\mathrm{tr}} = \frac{\delta^2}{n^2 \kappa^2} \mathcal{E}_{\mathrm{te}}. \tag{16}$$

The "≈" in 15 can be given several meanings. Firstly, it becomes an equivalence in an asymptotic limit in which $n$ and the number of eigenmodes in a given eigenvalue range (or the number of duplicate copies of any given eignemode) both grow large proportionally (Hastie et al., 2022; Bach, 2023). This is often phrased as sampling a proportional number of new eigenmodes from a fixed measure. Secondly, with fixed task eigenstructure, the error incurred can be bounded by a decaying function of $n$ (Cheng & Montanari, 2022). Thirdly, numerical experiments find small error even at quite modest $n$ (Canatar et al., 2021; Simon et al., 2021). For the purposes of this derivation, we will simply treat it as an equivalence.

---

[7] All the prior works cited at the start of the subsection find the same eigenframework. As of the time of writing, Cheng & Montanari (2022) give probably the most rigorous and general derivation for this eigenframework and could be taken as the canonical source if one is required.

[8] For two commuting matrices $\mathbf{A}, \mathbf{B}$, we will sometimes abuse notation slightly to write $\frac{\mathbf{A}}{\mathbf{B}}$ in place of $\mathbf{A}\mathbf{B}^{-1} = \mathbf{B}^{-1}\mathbf{A}$.

### E.2 REFRAMING RF REGRESSION AS KRR WITH STOCHASTIC EIGENSTRUCTURE.

We now turn to RF regression. Recall from the main text that RF regression is equivalent to KRR with the random feature kernel

$$\hat{K}(x, x') = \frac{1}{k} \sum_{j=1}^{k} g(w_i, x) g(w_i, x') \tag{17}$$

with $w_i$ sampled i.i.d. from some measure $\mu_w$. Recall also that there exists a spectral decomposition

$$g(w, x) = \sum_{i=1}^{\infty} \sqrt{\lambda_i} \phi_i(x) \zeta_i(w), \tag{18}$$

where $(\phi_i)_{i=1}^{\infty}$ and $(\zeta_i)_{i=1}^{\infty}$ are sets of eigenfunctions which are orthonormal over $\mu_x$ and $\mu_w$, respectively, and $(\lambda_i)_{i=1}^{\infty}$ are a decreasing sequence of nonnegative scalars. Note that the limiting kernel as $k \to \infty$ is $\lim_{k \to \infty} \hat{K}(x, x') = K(x, x') = \sum_i \lambda_i \phi_i(x) \phi_i(x')$ in probability, from which we see that we are indeed justified in reusing the notation $(\lambda_i, \phi_i)_{i=1}^{\infty}$ from the previous subsection.

Note that we can write this kernel as

$$\hat{K}(x, x') = \frac{1}{k} \sum_{i,i'=1}^{\infty} \sum_{j=1}^{k} \sqrt{\lambda_i \lambda_{i'}} \phi_i(x) \phi_{i'}(x) \zeta_i(w_j) \zeta_{i'}(w_j) \tag{19}$$

$$= \phi(x)^T \Lambda^{1/2} \frac{\mathbf{Z}\mathbf{Z}^T}{k} \Lambda^{1/2} \phi(x'), \tag{20}$$

where we define the projection matrix $(\mathbf{Z})_{ij} = \zeta_i(w_j)$. Comparing with Equation 13 and examining our KRR eigenframework, we see that, under Assumption A, we can predict the risk of RF regression as follows.

First, define

$$\tilde{\Lambda} := \Lambda^{1/2} \frac{\mathbf{Z}\mathbf{Z}^T}{k} \Lambda^{1/2}. \tag{21}$$

Then, let $\kappa$ be the unique nonnegative solution to the equation

$$n = \text{Tr}\left[\frac{\tilde{\Lambda}}{\tilde{\Lambda} + \kappa\mathbf{I}}\right] + \frac{\delta}{\kappa}. \tag{22}$$

Then test and train MSE will be well-approximated by

$$\mathcal{E}_{\text{te}} = \frac{n}{n - \text{Tr}\left[\left(\frac{\tilde{\Lambda}}{\tilde{\Lambda}+\kappa\mathbf{I}}\right)^2\right]} \left(\mathbf{v}^T \left(\frac{\kappa\mathbf{I}}{\tilde{\Lambda}+\kappa\mathbf{I}}\right)^2 \mathbf{v} + \sigma^2\right), \tag{23}$$

$$\mathcal{E}_{\text{tr}} = \frac{\delta^2}{n^2\kappa^2} \mathcal{E}_{\text{te}}. \tag{24}$$

We refer to this boxed set of equations as the *partially-evaluated RF eigenframework* because they are written in terms of the random projection $\mathbf{Z}$, which we still have to deal with.

### E.3 BUILDING UP SOME USEFUL STATISTICS OF $\tilde{\Lambda}$

The problem with the partially-evaluated RF eigenframework is of course that we do not know the stochastic eigenstructure matrix $\tilde{\Lambda}$. To make progress, we again turn to our Gaussian universality ansatz (Assumption A). Under this assumption, we may replace the columns of $\mathbf{Z}$ with i.i.d. isotropic Gaussian vectors, which amounts to replacing the whole of $\mathbf{Z}$ with i.i.d. samples from $\mathcal{N}(0, 1)$.

We now leverage a basic random matrix theory fact for such Gaussian matrices leveraged in many recent analyses of ridge regression with random design (Jacot et al., 2020a; Simon et al., 2021; Bach,

2023). First, let $\gamma \geq 0$ be the unique nonnegative solution to

$$k = \sum_i \frac{\lambda_i}{\lambda_i + \gamma} + \frac{k\kappa}{\gamma}. \tag{25}$$

The final term is $\frac{k\kappa}{\gamma}$ instead of simply $\frac{\kappa}{\gamma}$ as might be expected from the form of this fact in other works because of the factor of $\frac{1}{k}$ in Equation 21. Then, under the Gaussian design assumption on $\mathbf{Z}$, we have that

$$\mathbb{E}_{\mathbf{Z}}\left[\frac{\tilde{\mathbf{\Lambda}}}{\tilde{\mathbf{\Lambda}} + \kappa\mathbf{I}}\right] \approx \frac{\mathbf{\Lambda}}{\mathbf{\Lambda} + \gamma\mathbf{I}}. \tag{26}$$

This equation is useful because it reduces a statistic of the stochastic eigenstructure matrix $\tilde{\mathbf{\Lambda}}$ into a function of the known eigenvalue matrix $\mathbf{\Lambda}$.

**The meaning of "$\approx$."** The "$\approx$" in Equation 26 can be given various technical interpretations. It generally becomes an equivalence the *proportional limit* described in the following sense: consider fixing an integer $\eta > 1$ and increasing $n \to \eta n$, $k \to \eta k$, and also duplicating each eigenmode $\eta$ times. As $\eta \to \infty$, we reach the proportional limit. For the purposes of this derivation, we will simply treat it as an equivalence.

We now bootstrap this relation to obtain four more relations. We state these relations and then justify them.

$$\mathbb{E}_{\mathbf{Z}}\left[\frac{\kappa\mathbf{I}}{\tilde{\mathbf{\Lambda}} + \kappa\mathbf{I}}\right] \approx \frac{\gamma\mathbf{I}}{\mathbf{\Lambda} + \gamma\mathbf{I}}, \tag{27}$$

$$\mathbb{E}_{\mathbf{Z}}\left[\left(\frac{\tilde{\mathbf{\Lambda}}}{\tilde{\mathbf{\Lambda}} + \kappa\mathbf{I}}\right)^2\right] \approx \frac{\mathbf{\Lambda}}{\mathbf{\Lambda} + \gamma\mathbf{I}} - \frac{\kappa\mathbf{\Lambda}}{(\mathbf{\Lambda} + \gamma\mathbf{I})^2}\partial_\kappa\gamma, \tag{28}$$

$$\mathbb{E}_{\mathbf{Z}}\left[\frac{\kappa\tilde{\mathbf{\Lambda}}}{\left(\tilde{\mathbf{\Lambda}} + \kappa\mathbf{I}\right)^2}\right] \approx \frac{\kappa\mathbf{\Lambda}}{(\mathbf{\Lambda} + \gamma\mathbf{I})^2}\partial_\kappa\gamma, \tag{29}$$

$$\mathbb{E}_{\mathbf{Z}}\left[\left(\frac{\kappa\mathbf{I}}{\tilde{\mathbf{\Lambda}} + \kappa\mathbf{I}}\right)^2\right] \approx \frac{\gamma\mathbf{I}}{\mathbf{\Lambda} + \gamma\mathbf{I}} - \frac{\kappa\mathbf{\Lambda}}{(\mathbf{\Lambda} + \gamma\mathbf{I})^2}\partial_\kappa\gamma. \tag{30}$$

Taking a derivative of Equation 25 and performing some algebra, we have that

$$\partial_\kappa\gamma = \frac{k}{k - \sum_i \left(\frac{\lambda_i}{\lambda_i + \gamma}\right)^2}. \tag{31}$$

We obtain Equation 27 by simply subtracting both sides of Equation 26 from the identity matrix $\mathbf{I}$. We obtain Equation 29 by taking a derivative of Equation 26 with respect to $\kappa$. We obtain Equation 30 by taking a derivative of Equation 27 with respect to $\kappa$. Finally, we obtain Equation 28 from the identity $\left(\frac{\tilde{\mathbf{\Lambda}}}{\tilde{\mathbf{\Lambda}} + \kappa\mathbf{I}}\right)^2 = \mathbf{I} - 2\frac{\kappa\tilde{\mathbf{\Lambda}}}{(\tilde{\mathbf{\Lambda}} + \kappa\mathbf{I})^2} - \left(\frac{\kappa\mathbf{I}}{\tilde{\mathbf{\Lambda}} + \kappa\mathbf{I}}\right)^2$.

### E.4 INSERTING IDENTITIES INTO THE PARTIALLY-EVALUATED RF EIGENFRAMEWORK.

We are now in a position to insert Equations 26-30 into the partially-evaluated RF eigenframework to get closed-form results. We will generally trust that scalar quantities concentrate — that is, for some matrix $\mathbf{M}$ and vector $\mathbf{z}$ of interest, we will have that $\text{Tr}[\mathbf{M}] \approx \mathbb{E}[\text{Tr}[\mathbf{M}]]$ and $\mathbf{z}^T\mathbf{M}\mathbf{z} \approx \mathbb{E}[\mathbf{z}^T\mathbf{M}\mathbf{z}]$, with small enough error that we can neglect it.

We start with Equation 22 defining $\kappa$. Inserting Equation 26 into the trace, it becomes

$$n = \sum_i \frac{\lambda_i}{\lambda_i + \gamma} + \frac{\delta}{\kappa}. \tag{32}$$

Inserting Equations (28) and (30) into Equation 23, we get that

$$\mathcal{E}_{\text{te}} = \frac{n}{n - \sum_i \left(\frac{\lambda_i}{\lambda_i + \gamma} - \frac{\kappa\lambda_i}{(\lambda_i + \gamma)^2}\partial_\kappa\gamma\right)} \left(\sum_i \left(\frac{\gamma}{\lambda_i + \gamma} - \frac{\kappa\lambda_i}{(\lambda_i + \gamma)^2}\partial_\kappa\gamma\right)v_i^2 + \sigma^2\right). \tag{33}$$

Inserting Equation 31 for $\partial_\kappa \gamma$ and simplifying with the definitions $z = \sum_i \frac{\lambda_i}{\lambda_i + \gamma}$ and $q = \sum_i \left( \frac{\lambda_i}{\lambda_i + \gamma} \right)^2$ and the fact that $\kappa = \frac{\gamma}{k}(k - z)$ as per Equation 25, we arrive at the RF eigenframework we report in the main text.

**Remark on implicit constants.** The KRR eigenframework we started with had only one implicit constant $\kappa$, which could be understood two ways. First, it is given by the inverse of the trace of the inverse of the empirical kernel matrix: $\kappa \approx \mathrm{tr}[\boldsymbol{K}_{\mathcal{X}\mathcal{X}}^{-1}]^{-1}$ Wei et al. (2022). Second, it acts as an eigenvalue threshold: modes with $\lambda_i \gg \kappa$ are learned, and modes with $\lambda_i \ll \kappa$ are not. In the RF eigenframework, we have two implicit constants, $\kappa$ and $\gamma$. This new $\kappa$ still serves the first role: $\kappa \approx \mathrm{Tr}\left[ \hat{\boldsymbol{K}}_{\mathcal{X}\mathcal{X}}^{-1} \right]^{-1}$. However, it is now $\gamma$ that acts as the learnability threshold for eigenvalues.

### E.5 ADDITIONAL ESTIMATES: BIAS, VARIANCE, AND MEAN PREDICTOR

Test mean squared error is canonically split into a bias term (equal to the error of the "average" predictor) and a variance term (equal to the rest of the error). In the case of RF regression, a subtle question is: the average with respect to *what*? We could consider an average with respect to only random datasets, only random feature sets, or both at the same time. Jacot et al. (2020b) take a features-only average. Here we will take the other two.

In the main text, we denote the random dataset by $\mathcal{X}$. Let us also denote the random RF weights as $\mathcal{W}$. We then denote the *data-averaged bias and variance* to be

$$\mathtt{BIAS_d} := \mathbb{E}_{\mathcal{W}}\left[ \mathbb{E}_{x \sim \mu_x}\left[ \left( \mathbb{E}_{\mathcal{X}}[f(x)] - f_*(x) \right)^2 \right] \right] + \sigma^2, \tag{34}$$

$$\mathtt{VAR_d} := \mathbb{E}_{\mathcal{W}, \mathcal{X}}\left[ \mathbb{E}_{x \sim \mu_x}\left[ \left( f(x) - \mathbb{E}_{\mathcal{X}}[f(x)] \right)^2 \right] \right]. \tag{35}$$

Similarly, we let the *data-and-feature-averaged bias and variance* to be

$$\mathtt{BIAS_{d,f}} := \mathbb{E}_{x \sim \mu_x}\left[ \left( \mathbb{E}_{\mathcal{W}, \mathcal{X}}[f(x)] - f_*(x) \right)^2 \right] + \sigma^2, \tag{36}$$

$$\mathtt{VAR_{d,f}} := \mathbb{E}_{\mathcal{W}, \mathcal{X}}\left[ \mathbb{E}_{x \sim \mu_x}\left[ \left( f(x) - \mathbb{E}_{\mathcal{W}, \mathcal{X}}[f(x)] \right)^2 \right] \right]. \tag{37}$$

Fortunately, the KRR eigenframework gives us an equation for the data-averaged bias and variance: the data-averaged bias is the term in big parentheses in Equation 15, while the variance is the rest (see Simon et al. (2021)). This tells us that the data-averaged bias and variance are the following:

$$\mathtt{BIAS_d} \approx \mathcal{B}_d := \sum_i \left( \frac{\gamma}{\lambda_i + \gamma} - \frac{\kappa \lambda_i}{(\lambda_i + \gamma)^2} \frac{k}{k - q} \right) v_i^2 + \sigma^2, \tag{38}$$

$$\mathtt{VAR_d} \approx \mathcal{V}_d := \mathcal{E}_{\mathrm{te}} - \mathtt{BIAS_d}. \tag{39}$$

The more interesting case is perhaps the data-and-feature-averaged bias and variance. Jacot et al. (2020a); Canatar et al. (2021); Simon et al. (2021) found that the data-averaged predictor in the KRR case is simply $\mathbb{E}_{\mathcal{X}}[f(x)] = \boldsymbol{v}^T \frac{\boldsymbol{\Lambda}}{\boldsymbol{\Lambda} + \kappa \boldsymbol{I}} \boldsymbol{\phi}(x)$, so in our case it will be

$$\mathbb{E}_{\mathcal{X}}[f(x)] = \boldsymbol{v}^T \frac{\tilde{\boldsymbol{\Lambda}}}{\tilde{\boldsymbol{\Lambda}} + \kappa \boldsymbol{I}} \boldsymbol{\phi}(x). \tag{40}$$

Taking the feature average, we conclude that

$$\mathbb{E}_{\mathcal{W}, \mathcal{X}}[f(x)] = \mathbb{E}_{\mathcal{W}}\left[ \boldsymbol{v}^T \frac{\tilde{\boldsymbol{\Lambda}}}{\tilde{\boldsymbol{\Lambda}} + \kappa} \boldsymbol{\phi}(x) \right] = \boldsymbol{v}^T \frac{\boldsymbol{\Lambda}}{\boldsymbol{\Lambda} + \gamma \boldsymbol{I}} \boldsymbol{\phi}(x). \tag{41}$$

That is, $\mathbb{E}_{\mathcal{X}}[f(x)] \approx \sum_i \frac{\lambda_i}{\lambda_i + \gamma} v_i \phi_i(x)$. We thus conclude that the data-and-feature-averaged bias and variance are given as follows:

$$\mathtt{BIAS_{df}} \approx \mathcal{B}_{df} := \sum_i \left( \frac{\gamma}{\lambda_i + \gamma} \right)^2 v_i^2 + \sigma^2, \tag{42}$$

$$\mathtt{VAR_{df}} \approx \mathcal{V}_{df} := \mathcal{E}_{\mathrm{te}} - \mathtt{BIAS_{df}}. \tag{43}$$

### E.6 EVEN AT FIXED RIDGE, MORE IS BETTER FOR BIAS TERMS

Increasing $n$ and $k$ (and keeping $\delta$ constant) strictly decreases both data-averaged and data-and-feature-averaged bias. To see this, we fist note the following proposition, which follows from Equation 2:

**Proposition 1** (Derivatives of implicit constants). *Let $D := k\kappa\delta + (z-q)(k\kappa^2 + \gamma\delta) > 0$. Then we have that*

$$\partial_k \gamma = -\frac{\gamma(\gamma - \kappa)\delta}{D}, \tag{44}$$

$$\partial_k \kappa = \frac{\kappa^2(\gamma - \kappa)(z-q)}{D}, \tag{45}$$

$$\partial_n \gamma = -\frac{k\gamma\kappa^2}{D}, \tag{46}$$

$$\partial_n \kappa = -\frac{\kappa^2(k\kappa + (z-q)\gamma)}{D}. \tag{47}$$

In particular, note that when we increase $n$, we decrease both constants.[9] When we increase $k$, we decrease $\gamma$ but *increase $\kappa$*.

It is immediate from Equations (38) and (42) that increasing $k$ decreases both $\mathrm{BIAS_{df}}$ and $\mathrm{BIAS_{df}}$. It is immediate also that increasing $n$ decreases $\mathrm{BIAS_{df}}$, but since increasing $n$ decreases $\kappa$, it is not apparent that $\mathrm{BIAS_d}$ also decreases. However, we do not need to show this: it is apparent from Equation 68 that the bias of *KRR* is sample-wise monotonic in this sense for any task eigenstructure, and so RF regression (being simply a special case of KRR with stochastic task eigenstructure) will simply inherit this property. All together, we see that increasing either $n$ or $k$ will decrease both $\mathrm{BIAS_d}$ and $\mathrm{BIAS_{df}}$. Any region of increasing error one encounters when increasing $n$ or $k$ — for example, at a double-descent peak — can thus be pinned on (one or another notion of) the variance.

## F TAKING LIMITS OF THE RF EIGENFRAMEWORK

In Section 4, we report an omniscient risk estimator giving the expected test risk of RF regression in terms of task eigenstructure. Here we demonstrate that, by taking certain limits, we can recover several previously-reported results from our general framework. For easier reference, we repeat Equations 2 here:

$$n = \sum_i \frac{\lambda_i}{\lambda_i + \gamma} + \frac{\delta}{\kappa}, \tag{48}$$

$$k = \sum_i \frac{\lambda_i}{\lambda_i + \gamma} + \frac{k\kappa}{\gamma}. \tag{49}$$

### F.1 THE LIMIT OF LARGE $k$: RECOVERING THE KRR EIGENFRAMEWORK

RF regression converges to ordinary KRR in the limit of large $k$, and so we expect to recover the known KRR eigenframework. As $k \to \infty$, we find that $\frac{\kappa}{\gamma} \nearrow 1$. Therefore we can discard Equation 49 and find that $\kappa$ simply satisfies $n = \sum_i \frac{\lambda_i}{\lambda_i + \kappa} + \frac{\delta}{\kappa}$ as we get in the case of KRR.

Equation 3 reduces to

$$\mathcal{E}_{\mathrm{te}} = \frac{1}{1 - \frac{q}{n}} \left[ \sum_i \left( \frac{\kappa}{\lambda_i + \kappa} \right)^2 v_i^2 + \sigma^2 \right], \tag{50}$$

which is precisely the omniscient risk estimator for KRR (compare with e.g. Simon et al. (2021)).

---

[9]For this informal discussion, we will be a little fast and loose with terminology: when $\delta \to 0^+$, we will find that these partial derivatives may be zero instead of strictly negative, and so when we say e.g. that a derived quantity increases with $n$ or $k$, at zero ridge we really mean that it increases *or* remains the same.

## F.2 The limit of zero ridge: recovering the ridgeless framework of Bach (2023)

Bach (2023) report an omniscient risk estimator for ridgeless RF regression. We should recover this result from our framework when we take $\delta \to 0^+$. (Note that we cannot simply set $\delta = 0$ as Equations (48) and (49) do not always have solutions, but we will have no trouble taking the limit $\delta \to 0^+$.) Like Bach (2023), we will handle this in two cases.

**Case 1:** $n < k$. When $n < k$, we will still have $\kappa > 0$ even as $\delta \to 0^+$. Observe that $\gamma$ is determined by the constraint that $n = z = \sum_i \frac{\lambda_i}{\lambda_i + \gamma}$. Plugging into Equation 48 gives us that

$$k = n + \frac{k\kappa}{\gamma} \tag{51}$$

$$\Rightarrow \kappa = \frac{k-n}{k}\gamma. \tag{52}$$

Sticking in these substitutions into Equation 3 and simplifying substantially, we find that

$$\mathcal{E}_{\text{te}} = \frac{n}{n-q}\left[\sum_i \left(\frac{\gamma}{\lambda_i + \gamma}\right)^2 v_i^2 + \sigma^2\right] + \frac{n}{k-n}\left[\sum_i \frac{\gamma}{\lambda_i + \gamma}v_i^2 + \sigma^2\right], \tag{53}$$

which matches the result of Bach (2023).

**Case 2:** $n > k$. When $n > k$, we have that $\kappa \searrow 0$ as $\delta \to 0^+$. Therefore $\gamma$ is determined by the constraint that $k = \sum_i \frac{\lambda_i}{\lambda_i + \gamma}$. We also have that $k = z = \sum_i \frac{\lambda_i}{\lambda_i + \gamma}$. Inserting these facts into Equation 3, we find that

$$\mathcal{E}_{\text{te}} = \frac{1}{1 - \frac{k^2 - kq}{n(k-q)}}\left[\sum_i \frac{\gamma}{\lambda_i + \gamma}v_i^2 + \sigma^2\right] = \frac{n}{n-k}\left[\sum_i \frac{\gamma}{\lambda_i + \gamma}v_i^2 + \sigma^2\right]. \tag{54}$$

This also matches the result of Bach (2023).

**Remark.** We can observe the following proposition from the above $\delta \to 0^+$ liimt of the RF eigenframework:

**Proposition 2** (Monotonic improvement after the double-descent peak). *With $\delta \to 0^+$, we have that*

- $\frac{\partial \mathcal{E}_{\text{te}}}{\partial k} \leq 0$ *when $k > n$,*

- $\frac{\partial \mathcal{E}_{\text{te}}}{\partial n} \leq 0$ *when $n > k$.*

*Proof.* First, let us take $k > n$. From our discussion of "Case 1," we see that further increasing $k$ will leave $\gamma$ unchanged, which leaves $q$ unchanged. The only effect is thus to decrease the prefactor $\frac{n}{k-n}$ of the second term of Equation 53, which decreases $\mathcal{E}_{\text{te}}$.

Now let us take $n > k$. From our discussion of "Case 2," we see that further increasing $n$ leaves $\gamma$ unchanged. It is similarly immediate from Equation 54 that the result is to decrease $\mathcal{E}_{\text{te}}$. $\qquad\square$

Proposition 2 states that, at zero, ridge, increasing the larger of $n, k$ further can only improve test performance. Phrased another way, *it's all downhill after the double-descent peak.* Though this fact is elementary, we do not know of any existing proof in the literature.

## F.3 Student equals teacher: recovering the RF risk estimator of Maloney et al. (2022)

Maloney et al. (2022) work out a risk estimator for ridgeless RF regression under the "student equals teacher" condition that $\lambda_i = v_i^2$ and $\sigma^2 = \delta = 0$. Inserting these into Equations (53) and (54) and exploiting the fact that $z = \min(n, k)$ when $\delta = 0$ to simplify the resulting expressions, we find that

$$\mathcal{E}_{\text{te}} = \begin{cases} \frac{k}{k-n}n\gamma & \text{for } n < k, \\ \frac{n}{n-k}k\gamma & \text{for } n > k. \end{cases} \tag{55}$$

Again using the fact that $z = \min(n, k)$, these can be further unified (using the notation of Maloney et al. (2022)) as

$$\mathcal{E}_{\text{te}} = \begin{cases} \frac{k}{k-n}\Delta & \text{for } n < k, \\ \frac{n}{n-k}\Delta & \text{for } n > k, \end{cases} \tag{56}$$

where $\Delta \geq 0$ is the unique nonnegative solution to $1 = \sum_i \frac{\lambda_i}{m\lambda_i + \Delta}$ with $m := \min(n, k)$. This is the main result of Maloney et al. (2022).

### F.4 HIGH-DIMENSIONAL ISOTROPIC ASYMPTOTICS: THE SETTING OF MEI & MONTANARI (2019)

Here we compute the predictions of our framework in the setting of Mei & Montanari (2019), who study RF regression with isotropic covariates on the high-dimensional hypersphere. The essential feature of their setting is that task eigenvalues group into degenerate sets: we first have one eigenvalue of size $\Theta(1)$, then $d$ eigenvalues of size $\Theta(d^{-1})$, then $\Theta(d^2)$ eigenvalues of size $\Theta(d^{-2})$, and so on. We take $d\infty$ with $n/d \to r_{n/d}, k/d \to r_{k/d}$ with $r_{n/d}, r_{k/d} = \Theta(1)$. In this setting, we expect to perfectly learn the 0th-order modes, partially learn the 1st-order modes, and completely fail to learn the higher-order modes.

Let $\mathcal{I}_\ell$ denote the set of eigenvalue indices corresponding to degenerate level $\ell \geq 0$. Let $\mu_\ell = \sum_{i \in \mathcal{I}_\ell} \lambda_i$ be the total kernel power in level $\ell$ and $\mu_{\geq \ell}$ to denote $\sum_{m \geq \ell} \mu_\ell$. Let $p_\ell = \sum_{i \in \mathcal{I}_\ell} v_i^2$ be the total target power in level $\ell$ and $p_{\geq \ell}$ to denote $\sum_{m \geq \ell} p_\ell$. Let us write $\lambda_{\tilde{0}}, \lambda_{\tilde{1}}$, etc. to denote the value of the degenerate eigenvalue at level $\ell$.

In this setting, we will find that $\gamma, \kappa = \Theta(d^{-1})$. Equation 2 simplify in this setting to

$$\frac{n}{d} \approx \frac{\lambda_{\tilde{1}}}{\lambda_{\tilde{1}} + \gamma} + \frac{\delta + \mu_{\geq 2}}{d\kappa} \qquad \text{and} \qquad \frac{k}{d} \approx \frac{\lambda_{\tilde{1}}}{\lambda_{\tilde{1}} + \gamma} + \frac{k\kappa}{d\gamma}. \tag{57}$$

Here the $\approx$ hides $O(1)$ terms that asymptotically vanish. These equations can be solved analytically or numerically for $\gamma, \kappa$. Note that if one wishes to work in the large-$d$ limit, one might prefer to solve for $\tilde{\gamma} := d\gamma$ and $\tilde{\kappa} := d\kappa$ as follows:

$$r_{n/d} \approx \frac{\mu_1}{\mu_1 + \tilde{\gamma}} + \frac{\delta + \mu_{\geq 2}}{\tilde{\kappa}} \qquad \text{and} \qquad r_{k/d} \approx \frac{\mu_1}{\mu_1 + \tilde{\gamma}} + \frac{\tilde{\kappa}}{\tilde{\gamma}} r_{k/d}. \tag{58}$$

The advantage of the above equations is that all quantities are $\Theta(1)$ after one replaces $n/d, k/d$ with the appropriate $\Theta(1)$ ratios.

One then has $z = d\frac{\lambda_{\tilde{1}}}{\lambda_{\tilde{1}} + \gamma} = d\frac{\mu_1}{\mu_1 + \tilde{\gamma}}$ and $q = d\left(\frac{\lambda_{\tilde{1}}}{\lambda_{\tilde{1}} + \gamma}\right)^2 = d\left(\frac{\mu_1}{\mu_1 + \tilde{\gamma}}\right)^2$. Let $\tilde{z} := z/d = \Theta(1)$ and $\tilde{q} := q/d = \Theta(1)$. Equation 3 then reduces to

$$\mathcal{E}_{\text{te}} \approx \frac{1}{1 - \frac{\tilde{q}(1 - 2\tilde{z}) + \tilde{z}^2}{r_{n/d}(1 - \tilde{q})}} \left[ \left( \frac{\mu_1}{\mu_1 + \tilde{\gamma}} - \frac{\tilde{\kappa}\mu_1}{(\mu_1 + \tilde{\gamma})^2} \frac{1}{1 - \tilde{q}} \right) p_1 + p_{\geq 2} + \sigma^2 \right]. \tag{59}$$

We expect this is equivalent to the risk estimate of Mei & Montanari (2019)'s Definition 1, as they solve the same problem, but we have not directly confirmed equivalence.

### F.5 HIGH-DIMENSIONAL ISOTROPIC ASYMPTOTICS: THE SETTING OF MEI ET AL. (2022)

In followup work to Mei & Montanari (2019) in the linear regime, Mei et al. (2022) study RF regression in the same hyperspherical setting, but in a polynomial regime in which $n \propto d^a, k \propto d^b$ with $a, b > 0, a \neq b$, and $a, b \notin \mathbb{Z}$.[10]

In this regime, working through our equations, we find the following. First, let $c = \lfloor \min(a, b) \rfloor$. When $a > b$ and thus $n \gg k$, we find that $\gamma \approx \frac{\mu_{>c}}{k}$ and $\kappa \approx \frac{\delta}{n} \ll \gamma$. When $a < b$ and thus $k \gg n$, we find that $\gamma \approx \frac{\mu_{>c} + \delta}{n}$ and $\kappa \approx \gamma$. In either case, we find that $z, q \ll \min(k, n)$, and so the prefactor in Equation 3 becomes equal to 1, with the whole estimate simplifying to

$$\mathcal{E}_{\text{te}} \approx p_{>c} + \sigma^2. \tag{60}$$

That is, all signal of order $\ell \leq c$ is perfectly captured and all signal of order $\ell > c$ is fully missed (but not overfit), with $c$ set by the minimum of $n$ and $k$. This is precisely the conclusion of Mei et al. (2022).

### F.6 SANITY CHECK: THE LIMIT OF INFINITE RIDGE

It is easily verified that, as $\delta \to \infty$, we find that $\kappa, \gamma \to \infty$, and so $z, q \searrow 0$ and thus $\mathcal{E}_{\text{te}} = \sum_i v_i^2 + \sigma^2$ as expected.

---

[10]Technically they abstract the hyperspherical setting to a generic setting with a suitably gapped eigenspectrum, but the essential features are the same.

### F.7 INTERESTING CASE: THE LIMIT OF LARGE $n$

Here we report an additional limit which, to our knowledge, has not appeared in the literature. When $n \to \infty$ with $k$ finite, we have that $\kappa \searrow 0$ and thus $z \nearrow k$. The risk estimator then reduces to

$$\mathcal{E}_{\text{te}} = \sum_i \frac{\gamma}{\lambda_i + \gamma} v_i^2 + \sigma^2. \tag{61}$$

This resembles the bias term of the KRR risk estimator with $k$ samples, with the sole difference being the replacement $\left(\frac{\gamma}{\lambda_i + \gamma}\right)^2 \to \frac{\gamma}{\lambda_i + \gamma}$.

## G  PROOF OF THEOREM 1

*Proof.* First consider increasing $n \to n'$. Examining Equations 2, we see that we may always increase $\delta$ such that $\kappa$ and $\gamma$ remain unchanged and still satisfy both equations, and thus $z$ and $q$ are also unchanged. Turning to Equation 3, we see that increasing $n \to n'$ while keeping $\kappa, \gamma, k, z, q$ fixed can only decrease $\mathcal{E}_{\text{te}}$, as this only serves to decrease the (always positive) prefactor.

Next consider increasing $k \to k'$. From Equations 2, we see that it is always possible to increase $\delta$ so that $\gamma$ remains unchanged (and so $z, q$ remain unchanged) and $\kappa$ increases. Increasing $k$ decreases the prefactor in Equation 3 because $\frac{d}{dk} \frac{q(k-2z)+z^2}{n(k-q)} = -\frac{(z-q)^2}{(k-q)^2} \leq 0$, and increasing $k$ and $\kappa$ manifestly decreases the term in square brackets, and so overall $\mathcal{E}_{\text{te}}$ decreases.

To show that this inequality is strict in both cases, all we require is that $q > 0$, which only requires that the optimal $\delta$ is not positive infinity. To show this, we first observe that as $\delta \to \infty$, then $\kappa, \gamma$ grow proportionally, with $\frac{\kappa n}{\delta}, \frac{\gamma n}{\delta} \searrow 1$. It is then easy to expand Equation 3 in terms of large $\delta$, using the facts that $z = \gamma^{-1} \sum_i \lambda_i + O(\gamma^{-2})$ and $q = \gamma^2 + \sum_i \lambda_i^2 + O(\gamma^{-3})$. Inserting these expansions, we find that

$$\mathcal{E}_{\text{te}} = \left(1 + \gamma^{-2} n^{-2} \sum_i \lambda_i^2 + O(\gamma^{-3})\right) \left[\sum_i v_i^2 + \sigma^2 - 2\gamma^{-1} \sum_i \lambda_i v_i^2 + O(\gamma^{-2})\right] \tag{62}$$

$$= \lim_{\delta \to \infty} \mathcal{E}_{\text{te}} - 2\gamma^{-1} \sum_i \lambda_i v_i^2 + O(\gamma^{-2}). \tag{63}$$

From the negative $\gamma^{-1}$ term in this expansion, it is clear that the optimal $\gamma$ is finite so long as $\sum_i \lambda_i v_i^2 > 0$, and thus the inequality in the theorem is strict. $\qquad \square$

## H  NOISE SCALING AND COMPARISON WITH BENIGN OVERFITTING

Here we give further justification of our decision to scale the noise down with $n$ as $\sigma^2 = \sigma_{\text{rel}}^2 \cdot \mathcal{E}_{\text{te}}|_{\sigma^2=\delta=0}$, which amounts to $\sigma^2 \propto n^{-(\beta-1)}$. This turns out to be a subtle but very important point which differentiates our approach and conclusions from those of the "benign overfitting" literature (Bartlett et al., 2020; Tsigler & Bartlett, 2023). These other works take a fixed, positive noise level $\sigma^2 = \Theta(1)$. This is well-motivated — after all, there appears a priori to be no reason why the noise should change with $n$ — and has long been the standard setup in the statistics literature. One essential consequence of this setup — which we argue here is in fact somewhat misleading — is that, once $n$ grows large, the noise inevitably dominates the signal. Concretely, one can decompose the portion of test risk coming from the (deterministic) signal and that coming from the noise, and so long as the signal lies in the RKHS of the model and the ridge is not very large, at large $n$ the contribution to test error due to the noise dominates that coming from the signal.

We can see this clearly from the omniscient risk estimate for KRR which we use in the text. Examining Equation 65-Equation 68, it is easily seen that as $n$ grows, the implicit regularization $\kappa$ will decay to zero so long as $\delta = o(n)$ (for example, if it is constant). So long as the overfitting coefficient $\mathcal{E}_0$ does not explode as $n$ grows (and it usually does not, and will never if $\delta = \Theta(1)$), the contribution to $\mathcal{E}_{\text{te}}$ from the signal $\{v_i\}$ will decay to zero because $(1 - \mathcal{L}_i) = \frac{\kappa}{\lambda_i + \kappa} \to 0$. However, the contribution from the noise will still remain $\mathcal{E}_0 \sigma^2 = \Theta(1)$. Our foremost consideration in this case becomes guaranteeing that $\mathcal{E}_0 \to 1$. The esoteric eigenvalue decay $\lambda_i \propto i^{-1} \log^{-\gamma} i$ with $\gamma > 1$ found by Bartlett et al. (2020) achieves this, as shown by Mallinar et al. (2022).

Once we reach this "noise-dominating" regime, our task was irrelevant: to leading order, we might as well assume it is pure noise. This simplifies the question of overfitting dramatically and, we argue,

too much. Indeed, when training and testing on pure noise, the best one can hope to do is to fit as little as possible, because the Bayes-optimal predictor is uniformly zero. The optimal ridge is thus large, and zero ridge is suboptimal: the best we can hope for is that the suboptimality of zero ridge costs us only an $o_n(1)$ penalty in test error. That is, at best, interpolation is *permissible* (or "benign"); it is never *preferable*. This is emphatically not the regime we actually see in deep learning: one often finds that training to interpolation is often indeed strictly beneficial (practitioners wouldn't train for so many extra iterations if they weren't strictly helping!), and indeed Nakkiran & Bansal (2020); Mallinar et al. (2022) simply run experiments in which neural networks are fit to interpolation on data with controlled noise levels and find that interpolation is *not* benign but rather incurs a finite cost in test error. This suggests that we should seek theoretical settings where interpolation is beneficial on tasks that resemble real data but is harmful on pure noise, which is our contribution in Section 5.

As discussed above, the essential problem here is that finite noise inevitably grows to dominate uncaptured signal. The solution is to scale down noise proportional to the uncaptured signal as we do in Section 5. This puts both sources of error on an equal footing and enables us to make a nontrivial comparison between the two. As we show in Section 5, we do in fact need to consider the details of the target function to understand overfitting in modern machine learning (or at least KRR on image datasets), and so choosing a scaling that does not wash out all signal is crucial. Target-function-agnostic analyses like those performed by (Bartlett et al., 2020; Tsigler & Bartlett, 2023; Mallinar et al., 2022; Zhou et al., 2023) cannot resolve target-dependent effects.

It may seem aesthetically repugnant to have $\sigma^2$ scale in any way with $n$. The noise level is surely a fixed quantity; how can it change with the number of samples? In reality, this is simply a way of viewing whatever noise level we see in an experiment. As we increase $n$, we will indeed find that $\sigma_{\text{rel}}^2$ indeed increases (because the noise remains the same, but the uncaptured signal decays), but because we will never have infinite $n$ in practice, we will always observe a finite (or zero) value of $\sigma_{\text{rel}}^2$. At finite $n$, this $\sigma_{\text{rel}}^2$ may be large or small compared with the uncaptured signal. Our scaling boils down to a choice to treat $\sigma_{\text{rel}}^2$ as an order-unity quantity that one must consider carefully. The naive scaling with $\sigma^2 = \Theta(1)$ amounts to a choice to treat $\sigma_{\text{rel}}^2$ as infinite and dominating in the large-$n$ regime in which one proves theorems.

Since we are developing theory we wish to describe a set of real experiments, the choice of which scaling is preferable ought thus to be put to the empirical test. In our experiments, we find that $\sigma_{\text{rel}}^2$ is indeed small for the datasets we examine, which we view as affirming our choice to do theory in a regime in which the noise is not necessarily dominant.

Put another way, because we will always observe a finite $\sigma_{\text{rel}}^2$ in practice, we take a limit that *resembles* this situation, with signal and noise of the same order, but enables one to prove theorems as one can at large $n$. Figure 2 attests that the real experiments are well-described by our limit, and would be poorly-described by a limit which takes $\sigma_{\text{rel}}^2 \to \infty$.

We note that this business of how to scale quantities so effects of interest do not vanish is loosely similar to how the "neural tangent kernel" limit (Jacot et al., 2018) steadily loses feature learning as width grows, while the "$\mu$-parameterization" line of work (Yang & Hu, 2021) changes the layerwise scalings so feature change is no longer washed out by feature initialization. Yang et al. (2022); Vyas et al. (2023) observe that, when trying to understand a finite network using theory for infinite networks, one should scale up using the $\mu$-parameterization — not the neural tangent kernel parameterization — and this gives a (somewhat) theoretically-tractable infinite-width model which quantitatively captures the finite network one started with. Similarly, we find that scaling $\sigma^2$ in the manner we describe — and *not* keeping it $\Theta(1)$ — gives a tractable limit that closely resembles our experiments.

### H.1 RECOVERING BENIGN OVERFITTING FROM OUR RESULTS

That said, since we worked out our theory with $\sigma_{\text{rel}}^2$ as a free variable, we can always recover the benign overfitting "noise-dominated" case by simply taking setting $\sigma_{\text{rel}}^2 = \Theta(n^{\beta-1}) \to \infty$ post hoc, which gives $\sigma^2 = \Theta(1)$. (This will be nonrigorous, since some of the formal statements of Appendix I assumed $\sigma_{\text{rel}}^2 = \Theta(1)$ and would need to be reworked slightly, but this is a technicality, and we will find that the limit of large $\sigma_{\text{rel}}^2$ can be taken with no difficulty.)

**Test error in terms of fitting ratio with dominant noise.** When $\sigma_{\text{rel}}^2 \gg 1$, Lemma 10 reduces to approximately

$$\mathcal{E}_{\text{te}} \propto \frac{\sigma_{\text{rel}}^2}{1 + (\alpha - 1)\sqrt{\mathcal{R}_{\text{tr/te}}}}, \tag{64}$$

where we have neglected sub-leading-order error terms. Equation 64 tells us two useful facts. First, in our powerlaw setup, interpolation in this noise-dominated regime will *always* hurt us, specifically by a factor of $\alpha$. (This is consistent with the analysis of Mallinar et al. (2022), who find that the zero-ridge overfitting coefficient is $\mathcal{E}_0|_{\delta=0} = \alpha + o_n(1)$). Second, *as we take the limit $\alpha \to 1$, the additional "cost of interpolation" goes to zero, and we recover benign overfitting.* We now see that the peculiar eigenvalue decay $\lambda_i \propto i^{-1} \log^{-\gamma} i$ found to give benign overfitting is essentially a way to take the limit $\alpha \to 1$ without incurring the divergence in kernel norm one gets at exactly $\alpha = 1$.

Viewing such benign spectra as effectively having $\alpha \to 1$, we now see that as a consequence of our powerlaw analysis that for any target function with $\beta > 1$ and no noise, we expect zero ridge to be required for optimal fitting. We thus conjecture that ridge regression with the benign spectra already identified in the literature in fact exhibits *obligatory overfitting* for a wide set of reasonable target functions so long as there is no noise!

**Illustrating benign overfitting in our framework.** Figure 8 depicts the approach to benign overfitting as $\alpha \to 1$. Figure 9 shows how taking this limit actually makes overfitting obligatory if the target function is noiseless.

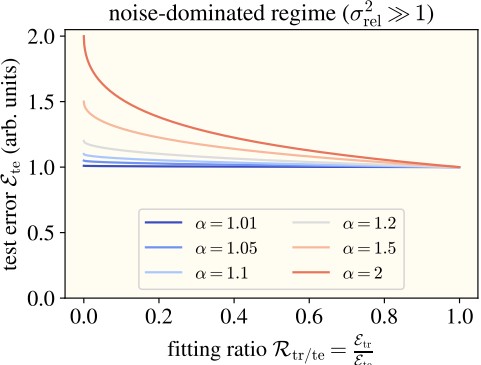

Figure 8: **When the target function is dominated by noise, taking $\alpha \to 1$ in our analysis recovers benign overfitting.** This plot shows theoretical fitting ratio vs. test error curves generated from Equation 64. As $\alpha$ approaches 1, it ceases to matter what fitting ratio one regularizes to: the test ratio is Bayes-optimal (up to sub-leading-order terms). Note the lack of resemblance between these theoretical noise-dominated curves and the experimental curves of Figure 2. Curves with $\alpha > 1$ illustrate "tempered overfitting" à la Mallinar et al. (2022).

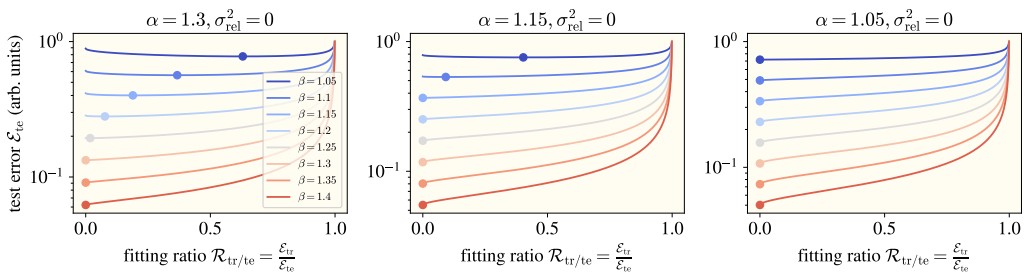

Figure 9: **The $\alpha \to 1$ eigenspectra known to exhibit benign overfitting in fact give *obligatory* overfitting on noiseless target functions.** This plot shows theoretical fitting ratio vs. test error curves assuming powerlaw eigenstructure and no noise, generated from Lemma 10. Each plot takes fixed eigenvalue exponent $\alpha$ and varying eigencoefficient exponent $\beta$. Dots show the location of the minimum of each curve. As per Corollary 1, interpolation ($\mathcal{R}_{\mathrm{tr/te}} = 0$) is optimal when $\beta \geq \alpha$. As $\alpha$ approaches 1 (plots left to right), interpolation becomes optimal for all target functions with powerlaw decay.

## H.2 RELATION TO "TEMPERED OVERFITTING"

Mallinar et al. (2022) describe a fitting behavior they term "tempered overfitting" in which training beyond optimal regularization to the point of interpolation incurs a penalty on test error which

is not negligible (as in benign overfitting) and is also not arbitrarily large, but rather takes the form of some multiplicative factor in $(1, \infty)$. In particular, they find that kernel regression with powerlaw eigenspectra $\lambda_i \sim i^{-\alpha}$ exhibits tempered overfitting in the noise-dominated regime, with the proportionality constant in fact equal to $\alpha$. This is reflected in Figure 8.

We find that these same spectra may exhibit obligatory overfitting when the targets also have powerlaw structure and the noise is not dominant. The task eigenstructures we consider here will generally be tempered as soon as interpolation is no longer optimal — that is, as soon as the condition of Corollary 1 is violated.

# I PROOFS: KRR WITH POWERLAW STRUCTURE

Here we will derive a picture of the train and test risk of KRR under powerlaw eigenstructure. The ultimate goal is to arrive at Theorem 2 giving the optimal fitting error ratio for exponents $\alpha$, $\beta$ and noise level. Along the way, we will derive closed-form equations for many quantities of interest, including the test risk $\mathcal{E}_{\text{te}}$ (Lemma 10), which are plotted as theory curves in Figure 2. All results in this appendix will assume powerlaw task eigenstructure as per Definition 2: that is, there exists some positive integer $i_0 = O(1)$ such that, for all $i \geq i_0$, it holds that $\lambda_i = i^{-\alpha}$ and $v_i^2 = i^{-\beta}$. (We do still assume that the eigenvalues are indexed in decreasing order, even though the first $i_0 - 1$ will not be exactly powerlaw.)

## I.1 RECALLING THE KRR EIGENFRAMEWORK

We begin by stating the $k \to \infty$ limit of the RF eigenframework of Section 4. In this limit, we recover a known risk estimate for KRR (or equivalently linear ridge regression) that has been converged upon by many authors in recent years (Sollich, 2001; Bordelon et al., 2020; Jacot et al., 2020a; Simon et al., 2021; Loureiro et al., 2021; Dobriban & Wager, 2018; Wu & Xu, 2020; Hastie et al., 2022; Richards et al., 2021) — a result which we actually bootstrapped to derive our RF eigenframework (Appendix E). We also recall this same eigenframework at the start of Appendix E. This risk estimate as follows:

Let $\kappa \geq 0$ be the unique nonnegative solution to

$$\sum_i \frac{\lambda_i}{\lambda_i + \kappa} + \frac{\delta}{\kappa} = n. \tag{65}$$

Then test risk is given approximately by

$$\text{MSE}_{\text{te}} \approx \mathcal{E}_{\text{te}} := \mathcal{E}_0 \mathcal{B}, \tag{66}$$

where the *overfitting coefficient* $\mathcal{E}_0$ is given by

$$\mathcal{E}_0 := \frac{n}{n - \sum_i \left(\frac{\lambda_i}{\lambda_i + \kappa}\right)^2} \tag{67}$$

and the *bias* is given by

$$\mathcal{B} = \sum_i \left(\frac{\kappa}{\lambda_i + \kappa}\right)^2 v_i^2 + \sigma^2. \tag{68}$$

As discussed in Appendix E, the "$\approx$" in 66 can be given several meanings. Firstly, it becomes an equivalence in an asymptotic limit in which $n$ and the number of eigenmodes in a given eigenvalue range (or the number of duplicate copies of any given eigenmode) both grow large proportionally (Hastie et al., 2022; Bach, 2023). This is often phrased as sampling a proportional number of new eigenmodes from a fixed measure. Secondly, with fixed task eigenstructure, the error incurred can be bounded by a decaying function of $n$ (Cheng & Montanari, 2022). Thirdly, numerical experiments find small error even at quite modest $n$ (Canatar et al., 2021; Simon et al., 2021). As with the RF eigenframework, in this paper we will simply treat it as an equivalence, formally proving facts about the risk estimate $\mathcal{E}_{\text{te}}$.

Recall that all sums run from $i = 1$ to $\infty$. Train risk is given by

$$\text{MSE}_{\text{tr}} \approx \mathcal{E}_{\text{tr}} := \frac{\delta^2}{n^2 \kappa^2} \mathcal{E}_{\text{te}}, \tag{69}$$

and so the fitting error ratio is given roughly by

$$\frac{\text{MSE}_{\text{tr}}}{\text{MSE}_{\text{te}}} \approx \mathcal{R}_{\text{tr/te}} := \frac{\mathcal{E}_{\text{tr}}}{\mathcal{E}_{\text{te}}} = \frac{\delta^2}{n^2 \kappa^2}. \tag{70}$$

Recall that the noise level is defined relative to the zero-noise-zero-ridge risk as

$$\sigma^2 = \sigma_{\text{rel}}^2 \cdot \mathcal{E}_{\text{te}}|_{\sigma^2 = \delta = 0}. \tag{71}$$

We will assume throughout that $\alpha > 1$ and $\beta \in (1, 2\alpha + 1)$. We will also assume $n \geq 1$. We will generally use an asterisk to demarcate quantities which occur at the optimal ridge w.r.t. test risk. For example, $\delta_* = \arg\min_\delta \mathcal{E}_{\text{te}}$, and $\kappa_* = \kappa|_{\delta=\delta_*}$, and $\mathcal{R}_{\text{tr/te}}^* = \mathcal{R}_{\text{tr/te}}|_{\delta=\delta_*}$.

It is worth noting that we have two varying parameters in our system: the sample size $n$ and the ridge $\delta$. Other quantities, like $\kappa$ or $\mathcal{R}_{\text{tr/te}}$, may be seen as functions of $n$ and $\delta$. When we state scaling results using big-$O$-style notation, they will describe *scaling with respect to $n$* — that is, $x = O(f(n, \delta))$ means that there exist constants $n_0, C > 0$ such that, for all $n \geq n_0$, it holds that $x \leq Cf(n, \delta)$. We will allow $\delta$ to vary arbitrarily, including as a function of $n$.

## I.2 CONTINUUM APPROXIMATIONS TO SUMS

Our main trick will be approximating the sums appearing in the eigenframework by integrals. The primary technical difficulty will be in bounding the error of these approximations (though we will ultimately find that these errors do not present a problem). Upon a first reading of this appendix, a reader may wish to simply ignore all error terms to grasp the overall flow of the argument.

Various quantities in this appendix will have complicated prefactors depending on the exponents $\alpha$ and $\beta$. These prefactors often cancel and simplify in the final accounting.[11] It is useful at first to pay less attention to these prefactors and more attention to how quantities scale — with respect to, for example, $n$ (which will be large) and $\kappa$ (which will be small).

Here we state continuum approximations to the three eigensums of the eigenframework.

**Lemma 1** (Continuum approximations to eigensums). *The sums appearing in the KRR eigenframework can be approximated as follows:*

$$\sum_{i=1}^{\infty} \frac{\lambda_i}{\lambda_i + \kappa} = \frac{\pi}{\alpha \sin(\pi/\alpha)} \kappa^{-1/\alpha} + O(1), \tag{72}$$

$$\sum_{i=1}^{\infty} \left( \frac{\lambda_i}{\lambda_i + \kappa} \right)^2 = \frac{\pi(\alpha - 1)}{\alpha^2 \sin(\pi/\alpha)} \kappa^{-1/\alpha} + O(1), \tag{73}$$

$$\sum_{i=1}^{\infty} \left( \frac{\kappa}{\lambda_i + \kappa} \right)^2 v_i^2 = \frac{\pi(\alpha - \beta + 1)}{\alpha^2 \sin\left( \pi \frac{(\beta-1)}{\alpha} \right)} \kappa^{\frac{\beta-1}{\alpha}} + O\left( \kappa^2 + \kappa^{\frac{\beta}{\alpha}} \right). \tag{74}$$

*Proof.* First, we argue that we may disregard the first $i_0 = O(1)$ terms in these sums and replace them with their ideal powerlaw values $\lambda_i = i^{-\alpha}$ and $v_i^2 = i^{-\beta}$. For the first two sums, note that replacing the first $i_0$ terms (or neglecting the first $i_0$ terms entirely) only incurs an $O(1)$ error, which is the size of the error terms in Equations (72) and (73) anyways. For the third sum, note that replacing the first $i_0$ terms (or neglecting the first $i_0$ terms entirely) only incurs an $O(\kappa^2)$ error, which is at most the size of the error term in Equation 74 anyways. We may thus proceed assuming perfect powerlaw structure, with $i_0 = 1$.

To prove the first clause, we note that the summand is monotonically decreasing, and thus use integrals to bound the sum as

$$\int_0^{\infty} \frac{i^{-\alpha}}{i^{-\alpha} + \kappa} \mathrm{d}i > \sum_{i=1}^{\infty} \frac{\lambda_i}{\lambda_i + \kappa} > \int_1^{\infty} \frac{i^{-\alpha}}{i^{-\alpha} + \kappa} \mathrm{d}i > \int_0^{\infty} \frac{i^{-\alpha}}{i^{-\alpha} + \kappa} \mathrm{d}i - 1. \tag{75}$$

The LHS integral evaluates to $\frac{\pi}{\alpha \sin(\pi/\alpha)} \kappa^{-1/\alpha}$, which is sufficient to give Equation 72.

Equation 73 is obtained in exactly the same way.

Equation 74 is obtained in the same way, except that the summand is no longer monotonically-decreasing if $\beta \in (2\alpha, 2\alpha + \beta)$, instead monotonically increasing to a maximum of size $\Theta(\kappa^{\beta/\alpha})$ at index $i_{\max} = \beta^{-1/\alpha}(2\alpha - \beta)^{1/\alpha}\kappa^{-1/\alpha} + O(1)$ before monotonically decreasing to zero. Splitting the sum into increasing and decreasing parts and again using integrals to bound the sum gives Equation 74. $\square$

**Remark.** We will more or less never again need to worry about the constant cutoff index $i_0$ and can forget about it now. Our conclusions will live in the regime of large $n$, and in this regime, it will be sufficient to have powerlaw tails, and we can simply neglect a constant number of eigenmodes at

---

[11]The authors are grateful for the existence of computer algebra systems.

low index. One should usually expect the sub-leading-order error terms we will carry around to be smaller the smaller $i_0$ is and the less the task deviates from perfect powerlaw eigenstructure, however (though one likely ought to consider some measure of total deviation rather than simply the cutoff $i_0$).

**Remark.** In several places in this appendix, including Equation 74, we will encounter the fraction $(\alpha - \beta + 1)/\sin(\pi(\beta - 1)/\alpha)$. This is nominally undefined when $\beta = \alpha + 1$. However, this discontinuity disappears with an application of L'Hopital's rule, simplifying to $\pi/\alpha$. (Indeed, when evaluating the integral to arrive at the RHS of Equation 74, if we specify that $\beta = \alpha + 1$, then e.g. Mathematica informs us that the integral evaluates to the result of applying L'Hopital's rule to the general form.) Rather than treat the case $\beta = \alpha + 1$ specially, we will simply gloss over it with the understanding that L'Hopital's rule ought to be used to resolve this undefined fraction.

### I.3 THE ZERO-NOISE-ZERO-RIDGE TEST RISK

Because the noise level is defined in a normalized fashion by $\sigma^2 = \sigma_{\text{rel}}^2 \cdot \mathcal{E}_{\text{te}}|_{\sigma^2=\delta=0}$, we need to find the zero-noise-zero-ridge risk $\mathcal{E}_{\text{te}}|_{\sigma^2=\delta=0}$ before we can study the noise in the general case. The zero-noise-zero-ridge risk is also an interesting object in its own right.

**Lemma 2** (Zero-ridge implicit regularization). *At $\delta = 0$, the implicit regularization $\kappa$ is given by*

$$\kappa|_{\delta=0} = \left( \frac{\pi}{\alpha \sin(\pi/\alpha)} \right)^{\alpha} n^{-\alpha} + O\left( n^{-(\alpha+1)} \right). \tag{76}$$

*Proof.* This follows straightforwardly from inserting Equation 72 into Equation 65. $\square$

**Lemma 3** (Zero-noise-zero-ridge test risk). *At $\sigma^2 = 0$ and $\delta = 0$, the test risk $\mathcal{E}_{\text{te}}$ is given by*

$$\mathcal{E}_{\text{te}}|_{\sigma^2=\delta=0} = \frac{\pi^{\beta}(\alpha - \beta + 1)}{\alpha^{\beta} \sin\left( \frac{\pi(\beta-1)}{\alpha} \right) (\sin(\pi/\alpha))^{\beta-1}} n^{-(\beta-1)} + O\left( n^{-2\alpha} + n^{-\beta} \right). \tag{77}$$

*Proof.* This follows from inserting Equation 76 into the continuum approximations of Lemma 1, then inserting the results into the the eigenframework to get $\mathcal{E}_{\text{te}}$, and finally simplifying. $\square$

**Remark.** In the process of doing this, one finds (after a surprising cancellation) that the overfitting coefficient is simply $\mathcal{E}_0|_{\delta=0} = \alpha + O(n^{-1})$, as previously reported by Mallinar et al. (2022).

### I.4 TEST RISK IN TERMS OF $\kappa$

We now turn back to the general case in which noise and regularization are nonzero. The following lemma gives test risk in terms of the implicit regularization $\kappa$.

**Lemma 4.**
$$\mathcal{E}_{\text{te}} = \frac{n}{n - \frac{\pi(\alpha-1)}{\alpha^2 \sin(\pi/\alpha)} \kappa^{-1/\alpha}}$$
$$\times \left( \frac{\pi(\alpha - \beta + 1)}{\alpha^2 \sin\left( \pi \frac{(\beta-1)}{\alpha} \right)} \kappa^{\frac{\beta-1}{\alpha}} + \sigma_{\text{rel}}^2 \cdot \frac{\pi^{\beta}(\alpha - \beta + 1)}{\alpha^{\beta} \sin\left( \frac{\pi(\beta-1)}{\alpha} \right) (\sin(\pi/\alpha))^{\beta-1}} \cdot n^{-(\beta-1)} \right)$$
$$+ O(n^{-\beta} + \kappa^2 + \kappa^{\beta/\alpha}). \tag{78}$$

*Proof.* This lemma follows from inserting Lemmas 1 and 3 into the definition of $\mathcal{E}_{\text{te}}$. $\square$

**Lemma 5.** $\kappa = \Omega\left( n^{-\alpha} \right), \mathcal{E}_{\text{te}} = \Omega\left( \min\left( \kappa^{\frac{\beta-1}{\alpha}}, 1 \right) \right),$ *and thus* $\mathcal{E}_{\text{te}} = \Omega\left( n^{-(\beta-1)} \right).$

*Proof.* The fact that $\kappa = \Omega\left( n^{-\alpha} \right)$ follows from the fact that $\kappa|_{\delta=0} = \Theta(n^{-\alpha})$ and $\kappa$ is a monotonically increasing function of $\delta$.

To see that $\mathcal{E}_{\text{te}} = \Omega\left( \min\left( \kappa^{\frac{\beta-1}{\alpha}}, 1 \right) \right)$, we return to the definition of $\mathcal{E}_{\text{te}}$. It is easily seen that $\mathcal{E}_0 = \Theta(1)$, because $\mathcal{E}_0|_{\delta=0} = \Theta(1)$, $\mathcal{E}_0 \geq 1$, and $\mathcal{E}_0$ monotonically decreases with $\delta$, so we need only examine the bias $\mathcal{B}$ to understand the size of $\mathcal{E}_{\text{te}}$ in a scaling sense. Because $\sigma_{\text{rel}}^2 = O(1)$, we have that $\sigma^2 = \sigma_{\text{rel}}^2 \cdot \mathcal{E}_{\text{te}}|_{\sigma^2=\delta=0} = O(n^{-(\beta-1)})$. Examining the eigensum $\sum_i (1 - \mathcal{L}_i)^2 v_i^2$, it is easily seen that at all modes $i > i'$ for some $i' = \Theta(\max(1, \kappa^{-1/\alpha}))$, we will have that $\mathcal{L}_i = \frac{i^{-\alpha}}{i^{-\alpha}+\kappa} \leq \frac{1}{2}$,

say, which tells us that $\mathcal{E}_{\text{te}} = \Omega(\int_{i'}^{\infty} i^{-\beta} di) = \Omega\left((i')^{-(\beta-1)}\right)$[12]. The second clause of the lemma follows from this.

The third clause of the lemma follows from the insertion of the first statement into the second. □

**Lemma 6.** *At optimal regularization, we have that* $\mathcal{E}_{\text{te}}^* = \Theta(n^{-(\beta-1)})$, $\kappa_* = \Theta(n^{-\alpha})$, *and thus* $\delta_* = O(n^{-(\alpha-1)})$

*Proof.* The first clause of this lemma follows from the lower bound on $\mathcal{E}_{\text{te}}$ given by Lemma 5 and the fact that we can saturate this lower bound in a scaling sense because we do so already at zero ridge, $\mathcal{E}_{\text{te}}|_{\delta=0} = \Theta(n^{-(\beta-1)})$ (which itself follows from inserting Lemma 2 into Lemma 4). Given this, the second clause of Lemma 5 assures us that $\kappa_* = \Theta(n^{-\alpha})$ as desired.[13] The bound on $\delta_*$ follows from Equation 72 and Equation 65. □

**Remark.** Lemma 5 is useful because it tells us that, at optimal ridge, the error terms in Lemma 4 will indeed decay faster with $n$ than $\mathcal{E}_{\text{te}}$ itself.

### I.5 RELATING $\kappa$ AND $\mathcal{R}_{\text{tr/te}}$

**Lemma 7.** *The fitting ratio is given in terms of $\kappa$ by*

$$\sqrt{\mathcal{R}_{\text{tr/te}}} = 1 - n^{-1} \frac{\pi}{\alpha \sin(\pi/\alpha)} \kappa^{-1/\alpha} + O(n^{-1}). \tag{79}$$

*Solving for $\kappa$, we have*

$$\kappa = \left(\frac{\pi}{\alpha \sin(\pi/\alpha)}\right)^{-\alpha} n^{\alpha} \left(1 - \sqrt{\mathcal{R}_{\text{tr/te}}} + O(n^{-1})\right)^{-\alpha}. \tag{80}$$

*Proof.* This lemma follows directly from Equations (65), (70) and (72). □

**Remark.** A minor annoyance in the usage of Equation 80 is that, when $\mathcal{R}_{\text{tr/te}}$ is too close to 1, the $O(n^{-1})$ term can affect $\kappa$ to leading order. As a patch to avoid this, we will initially restrict the domain of study to $\mathcal{R}_{\text{tr/te}} \in [0, 1-c]$ for some $n$-independent constant $c > 0$. We are then assured that

$$\kappa = \left(\frac{\pi}{\alpha \sin(\pi/\alpha)}\right)^{-\alpha} n^{\alpha} \left(1 - \sqrt{\mathcal{R}_{\text{tr/te}}}\right)^{-\alpha} + O(n^{-(\alpha+1)}). \tag{81}$$

The fact that $\kappa$ increases monotonically with $\mathcal{R}_{\text{tr/te}}$ (Lemma 8) will let us extend results of interest to the small region $\mathcal{R}_{\text{tr/te}} \in (1-c, 1)$. As far as the study of the optimal regularization point go, we need not consider this small edge region in the following sense:

**Lemma 8.** *The fitting error ratio $\mathcal{R}_{\text{tr/te}} \in [0, 1)$ and implicit regularization $\kappa \in [\kappa|_{\delta=0}, \infty)$ are monotonically-increasing functions of each other.*

*Proof.* This lemma is apparent from the fact that $\mathcal{R}_{\text{tr/te}} = \left(1 - n^{-1} \sum_i \frac{\lambda_i}{\lambda_i + \kappa}\right)^2$. □

**Lemma 9.** *The optimal value of the fitting error ratio is bounded away from 1 in the sense that there exists a constant $c' > 0$ such that $\mathcal{R}_{\text{tr/te}}^* \leq 1 - c' + O(n^{-1})$.*

*Proof.* We know from Lemma 6 that $\kappa_* = \Theta(n^{-\alpha})$. We then obtain the desired result from Equation 79. □

### I.6 PUTTING IT ALL TOGETHER: TEST RISK IN TERMS OF $\mathcal{R}_{\text{tr/te}}$

The following lemma gives the test risk $\mathcal{E}_{\text{te}}$ in terms of the fitting error ratio $\mathcal{R}_{\text{tr/te}}$ and is the basis for our ultimate conclusions about the optimal fitting error ratio.

---

[12]It might seem that we need to be mindful of the cutoff index $i_0$ up to which we do not necessarily have powerlaw eigenstructure, but we do not actually: because $i_0 = \Theta(1)$, we will still have $i' = \Theta(\max(1, \kappa^{-1/\alpha}))$.

[13]Seeing this is made easier by writing out the scaling notation explicitly — e.g., $\mathcal{E}_{\text{te}} = \Omega(n^{-(\beta-1)})$ means that there exists a constant $A_1 > 0$ such that $\mathcal{E}_{\text{te}} >_1 \cdot n^{-(\beta-1)}$ for sufficiently large $n$, and so on — and then solving for constants which are sufficient to give the desired new statements.

**Lemma 10.** *Let $c > 0$ be an $n$-independent constant. Then for $\mathcal{R}_{\mathrm{tr/te}} \in [0, 1-c]$, we have that*

$$
\mathcal{E}_{\mathrm{te}} = \pi^\beta \alpha^{-\beta} (\alpha - \beta + 1) \left( \sin\left(\frac{\pi}{\alpha}\right) \right)^{-(\beta-1)} \csc\left(\frac{\pi(\beta-1)}{\alpha}\right)
$$

$$
\times \frac{1}{1 + (\alpha - 1)\sqrt{\mathcal{R}_{\mathrm{tr/te}}}} \left( \alpha \sigma_{\mathrm{rel}}^2 + \left(1 - \sqrt{\mathcal{R}_{\mathrm{tr/te}}}\right)^{-(\beta-1)} \right) n^{-(\beta-1)}
$$

$$
+ O(n^{-\beta} + n^{-2\alpha}), \quad (82)
$$

*where the suppressed constants in the $O(n^{-\beta} + n^{-2\alpha})$ term may depend on $c$.*

*Proof.* This lemma follows from insertion of Equation 81 into Lemma 4. $\square$

**Lemma 11.** *Let us abbreviate $r = \sqrt{\mathcal{R}_{\mathrm{tr/te}}}$, and let $c$ be the $n$-independent constant from Lemma 10. Then for $\mathcal{R}_{\mathrm{tr/te}} \in [0, 1-c]$, we have that the second derivative of the log of the train error obeys*

$$
\frac{d^2}{dr^2} \log \mathcal{E}_{\mathrm{te}} = \frac{(\alpha - 1)^2}{(1 + (1-\alpha)r)^2} + \frac{\beta - 1 + \alpha\beta(\beta-1)(1-r)^{\beta-1}\sigma_{\mathrm{rel}}^2}{(1 - r + \alpha(1-r)^\beta \sigma_{\mathrm{rel}}^2)^2} + O(n^{-1} + n^{-2\alpha+\beta-1})
$$

$$
(83)
$$

$$
\geq \frac{(\alpha - 1)^2}{\alpha^2} + O(n^{-1} + n^{-2\alpha+\beta-1}), \quad (84)
$$

*and thus at sufficiently large $n$, $\mathcal{E}_{\mathrm{te}}$ is strongly logarithmically convex w.r.t. $r$.*

*Proof.* This lemma follows from taking two derivatives of $\mathcal{E}_{\mathrm{te}}$ as given by Lemma 10. $\square$

**Lemma 12.** *For all $\mathcal{R}_{\mathrm{tr/te}} \in [0, 1)$, we have that*

$$
\frac{\mathcal{E}_{\mathrm{te}}}{\mathcal{E}_{\mathrm{te}}^*} \geq 1 + \frac{(\alpha - 1)^2}{2\alpha^2} \left( \sqrt{\mathcal{R}_{\mathrm{tr/te}}} - \sqrt{\mathcal{R}_{\mathrm{tr/te}}^*} \right)^2 + O(n^{-1} + n^{-2\alpha+\beta-1}). \quad (85)
$$

*Proof.* It is easy to see that this holds for all $\mathcal{R}_{\mathrm{tr/te}} \in [0, 1-c]$ for any constant $c > 0$. Indeed, it follows directly from Lemma 11. Note that

$$
\log \mathcal{E}_{\mathrm{te}} - \log \mathcal{E}_{\mathrm{te}}^* \geq \frac{(\alpha-1)^2}{2\alpha^2} \left( \sqrt{\mathcal{R}_{\mathrm{tr/te}}} - \sqrt{\mathcal{R}_{\mathrm{tr/te}}^*} \right)^2 + O(n^{-1} + n^{-2\alpha+\beta-1}), \quad (86)
$$

so

$$
\frac{\mathcal{E}_{\mathrm{te}}}{\mathcal{E}_{\mathrm{te}}^*} \geq \exp\left\{ \frac{(\alpha-1)^2}{2\alpha^2} \left( \sqrt{\mathcal{R}_{\mathrm{tr/te}}} - \sqrt{\mathcal{R}_{\mathrm{tr/te}}^*} \right)^2 \right\} + O(n^{-1} + n^{-2\alpha+\beta-1}), \quad (87)
$$

which yields Equation 85 using the fact that $e^x > 1 + x$ for $x \in \mathbb{R}$. We now need to assure ourselves that this bound still holds for $\mathcal{R}_{\mathrm{tr/te}} \in (1 - c, 1)$. Intuitively, the test tisk will generally explode as $\mathcal{R}_{\mathrm{tr/te}} \to 1$, growing from $\Theta(n^{-(\beta-1)})$ to $\Theta(1)$, so we should expect this lower bound to hold near $\mathcal{R}_{\mathrm{tr/te}} = 1$ with little trouble. Since the error terms we have been carrying around grow large in this region, we will need to rely on Lemmas 5 and 8 for support.

First, observe that there exist constants $B_1$, $n_1$ such that, if

$$
\mathcal{E}_{\mathrm{te}} \geq B_1 n^{-(\beta-1)} \qquad \text{for all } n \geq n_1 \quad (88)
$$

on the region $\mathcal{R}_{\mathrm{tr/te}} \in (1 - c, 1)$, then Equation 85 will hold on this region. (For example, one might set $\mathcal{R}_{\mathrm{tr/te}} = 1$ on the RHS of Equation 85, then use the fact that $\mathcal{E}_{\mathrm{te}}^* = \Theta(n^{-(\beta-1)})$ (Lemma 6), then increase the constant a small amount to absorb the $O(n^{-1} + n^{-2\alpha+\beta-1})$ error term. ) Then note that, by Lemma 5, there exist constants $B_2$, $n_2$ such that

$$
\mathcal{E}_{\mathrm{te}} \geq B_2 \kappa^{-(\beta-1)/\alpha} \qquad \text{for all } n \geq n_2. \quad (89)
$$

Thus, provided that $n > \max(n_1, n_2)$, we are assured of Equation 88 so long as $c$ is small enough that $\kappa \geq (B_2/B_1)^{\alpha/(\beta-1)} n^{-\alpha}$. We can identify such a sufficiently small $c$ by looking at Equation 79, which tells us that so long as $n \geq n_3$ for some $n_3$, we can be assured that $\kappa$ is sufficiently large when $\mathcal{R}_{\mathrm{tr/te}} = 1 - c$ for $c = (B_1/B_2)^{1-\beta} \pi/(\alpha \sin(\pi/\alpha))$. The fact that $\kappa$ monotonically increases with $\mathcal{R}_{\mathrm{tr/te}}$ (Lemma 8) assures us that $\kappa$ will remain sufficiently large for all $\mathcal{R}_{\mathrm{tr/te}} \in (1 - c, 1)$, and thus Equation 88 will continue to hold. This patches over our edge case and completes the proof. $\square$

**Lemma 13.** *The value of the fitting error ratio that minimizes the test error is*

$$\mathcal{R}^*_{\text{tr/te}} = r^2_* + O(n^{-1} + n^{-2\alpha+\beta-1}), \tag{90}$$

*where $r_*$ is either the unique solution to*

$$\alpha - \beta - (\alpha - 1)\beta r + \alpha(\alpha - 1)(1 - r_*)^\beta \sigma^2_{\text{rel}} = 0 \tag{91}$$

*over $r_* \in [0, 1)$ or else zero if no such solution exists.*

*Proof.* First, let the constant $c$ in Lemma 10 be less than $c'/2$, where $c'$ is the constant prescribed by Lemma 9, so that we are assured that $\mathcal{R}^*_{\text{tr/te}} \in [0, 1 - 2c]$. Because the error terms are small relative to the quantity itself in the region $\mathcal{R}_{\text{tr/te}} \in [0, 1 - c]$, we may simply take a derivative of $\mathcal{E}_{\text{te}}$ in terms of $\mathcal{R}_{\text{tr/te}}$ as given by Lemma 10, setting $\frac{d\mathcal{E}_{\text{te}}}{dr} = 0$. This yields Equations (90) and (91). We are assured that this equation can have only one solution on the domain of interest because the function we differentiated to obtain it is strongly log-convex for $r \in [0, 1)$, as shown in proving Lemma 11. If this equation has no solution, this implies that there must be no local minimum on the domain, so the minimum over the domain must lie at (or more precisely, because we have error terms, *close to*) an endpoint — that is, at either $\mathcal{R}_{\text{tr/te}} = 0 + O(n^{-\gamma})$ or $\mathcal{R}_{\text{tr/te}} = 1 - c + O(n^{-\gamma})$, where $\gamma = \min(1, 2\alpha - \beta + 1)$. However, we chose $c$ to be small enough that $\mathcal{R}_{\text{tr/te}} < 1 - 2c + O(n^{-\gamma})$, so we can eliminate the right endpoint, and we have that $\mathcal{R}^*_{\text{tr/te}} = O(n^{-\gamma})$ as desired. $\qquad \square$

**Remark.** It is worth emphasizing that, because we were free from the beginning to choose $c$ to be quite small, the rather technical patch business in the preceding proof is not really all that important and can be glossed over on a first reading. It is, however, nice to have for completeness, as covering the whole region $\mathcal{R}_{\text{tr/te}} \in [0, 1)$ permits the final theorem statement to be simpler.

### I.7 STATING THE FINAL THEOREM

Putting together Lemmas 12 and 13 gives us Theorem 2.

