# OpenReview forum: "More is Better: when Infinite Overparameterization is Optimal and Overfitting is Obligatory"
_ICLR.cc/2024/Conference — ICLR 2024 poster_

### Official Review · Reviewer_cEFV · 2023-10-29

**Soundness:** 2 fair
**Presentation:** 3 good
**Contribution:** 2 fair
**Rating:** 6
**Confidence:** 2

**Summary:**

This paper aims to provide theoretical backing for the recent empirical study about the scaling "law" of large language models, i.e. larger model size + more data + more compute = more performance. To this end, the authors propose to study the random-feature (RF) regression model and show that their approximation to the test error strictly decreases when more data points and more width are provided. Then, they characterize which tasks obtain optimal performance when the training error is much smaller than the test error. Finally, some empirical evidence supporting the theory is provided.

**Strengths:**

The paper is very well written and polished. I believe the idea is original, although I'm not very familiar with this field beyond the standard NTK/NNGP theory (see my question below, though)---my confidence reflects this. The presentation is in general, clear; although some specific parts can be improved. More importantly, the paper's aim to formalize the common beliefs in the field---in what people call scaling "law"---is laudable.

**Weaknesses:**

First, as I mentioned before, in some parts, the presentation can be improved. Here is a non-exhaustive list:

1. How are the approximations of the train and test errors on page 4 derived. I think it is crucial to at least provide intuitions about them and quantify the approximation errors. This is because much of the results of the paper rely on those approximations.
2. The term "interpolation" keeps coming up without being defined. Maybe it is obvious for people in this specific field; however, for the benefit of a broader audience, the authors should be more verbose.

Second, I'm unsure about the significance of the theoretical results. Mainly because the authors claim that the test error becomes better as $n$, $k$ increase because the RF model converges to the kernel regression limit. However, it is a bit hard to believe since kernel machines (RFs, GPs) are in general worse than their finite-width NNs counterpart (kernel machines = linear models => no representation learning). As the authors mentioned on page 5, Thm. 1 works since intuitively RF -> kernel machine; the latter is the ideal limiting process. If this is the case, doesn't this imply that GPs should be better than standard NNs?

**Questions:**

1. Could the authors please address the above concerns?
2. The authors mentioned that "we do not want to take a fixed noise variance" and then argued that it is better to scale it proportional to the uncaptured signal. I'm not sure if this is realistic since $\sigma^2$ is the aleatoric (irreducible) uncertainty inherent in the data (think about the noise resulting from some measurement error in a sensor). How do you handle this, then? Maybe you should instead argue using the language of GPs, and thus give you the freedom to talk about _epistemic uncertainty_, which decreases as $n$ increases?
3. How should one relate Thm. 1 and Fig. 1? In the former, the test error should always decrease, but in the latter, the test error saturates. Note that, I think Fig. 1 is the "correct" behavior: kernel machines are not powerful and thus the test error is bounded away from zero, higher than DNN's. (As for the train error, this seems fine since infinite-width => linearity => quadratic loss => global optimum is achievable.)
4. How does this paper relate to the work of [Bowman & Montufar (NeurIPS 2022)](https://arxiv.org/abs/2206.02927), which also studies the large-width large-data regime?

Thanks!

---

> ### Author Response · Authors · 2023-11-15
> **Author response to weaknesses and questions**
>
> We thank the reviewer for their evaluation, noting in particular the clear writing and effort to crystallize common wisdom. To the first weakness: we opted for a streamlined, nonrigorous approach for our derivation of the RF eigenframework, adopting the philosophy typical in statistical physics where one makes justified approximations and then checks against numerics at the end of the calculation. As we discuss at various points, we expect our RF eigenframework to become exact in a proportional limit (e.g. precisely the limit taken by Bach et al (2023)). Extracting approximation errors for finite $n, k$ is not a trivial task, but we know of other work in progress that intends to do this.
>
> To the second weakness: thank you for flagging this. We have added a few helpful phrases to help the reader understand that by “interpolation” we mean “zero training error.”
>
> To the question about significance: we find that [the limiting KRR] is better than [finite RF regression]. However, this does not mean that KRR is arbitrarily good! Some other model type (e.g., a neural network) can certainly be better, and we make no claims about performance relative to any model outside of RF regression. To further clarify the relationship between these model classes: RF regression trains only the last layer, does not learn features, and becomes KRR in the limit of infinitely-many features. Infinite-width NNs are a special case of KRR and do not learn features, so long as one scales init + lr in the NTK fashion. Finite-width NNs learn features and are not simply related to RF regression or KRR. Heuristically speaking, we show that infinite-width NNs are better than RF regression, but whether infinite-width NNs are better than finite-width NNs remains an open problem.
>
> To the enumerated questions:
> 1. We hope we have addressed your concerns. We are happy to offer additional clarifications.
> 2. We refer the reviewer to Appendix E discussing the motivation for scaling down noise with test error. Ultimately, this is just a theoretical convenience: in practice, $\sigma^2$ will indeed be whatever finite quantity it actually is, and $n$ will be finite. For the purposes of theoretical calculation, we want to take $n$ to infinity, and so we’re left with the question of what to do with $\sigma^2$ to preserve behavior similar to what we started with. We argue that keeping $\sigma^2$ unchanged as $n \rightarrow \infty$ actually changes the behavior in undesirable ways, and one instead ought to scale it down in the manner described. This is just a trick for writing down a better model for our learning task and does not actually amount to an assumption that the noise is really $n$-dependent.
> 3. Ah, there is no conflict: in Figure 1, the test error decreases continuously, and it *also* approaches an asymptote. Consider, for example, the function $x^{-1} + 1$ as $x$ grows: it always decreases, yet never decreases below its asymptote of $1$. Theorem 1 only promises that the gains from increasing $k$ are positive, but they will become smaller and smaller as $k$ grows.
> 4. Like us, Bowman and Montufár (2022) compare the learning of models with finite sample size and width to those of infinite sample size and width. They focus on training dynamics: their claim is of the flavor that “even at finite sample size and width, the evolution of the learned function looks like the infinite case early in training.” This is a similar message to the (empirical) message of the “[deep bootstrap](https://arxiv.org/abs/2010.08127).” By contrast, we study generalization rather than the training trajectory, focus on the learned function at the end of training rather than early in training.
>
> We thank the reviewer again for their review. If they feel we have addressed some of their concerns, we ask that they consider adjusting their score accordingly.

---

> > ### Comment · Reviewer_cEFV · 2023-11-20
> > **Reply**
> >
> > Thanks for the clarification! I have no further comments and updated my score accordingly.

---

> > > ### Author Response · Authors · 2023-11-21
> > >
> > > Much appreciated :)

---

### Official Review · Reviewer_S1iV · 2023-10-31

**Soundness:** 4 excellent
**Presentation:** 3 good
**Contribution:** 4 excellent
**Rating:** 8
**Confidence:** 3

**Summary:**

The paper studies how scaling the model size and the dataset size affects the performance of random features models.
While "classical" learning theory predicts that scaling the model can hurt performance, causing the model to overfit the data, this seems not to be true when learning with modern tools, e.g. neural networks. In practice, it seems that the bigger the model is, the better the performance it gets.
This paper recovers this behavior in the setting of learning ridge regression with random features. Namely, the authors establish that, for a correctly tuned ridge regression, the model's test error always improves with the number of features and the number of training samples. That is, the "double descent" phenomenon can be eliminated by correctly tuning the model's parameters. Additionally, the authors show that overfitting is necessary to achieve good performance in some settings with low noise. That is, not only overfitting doesn't hurt performance, but it also enables reaching optimal test error.

**Strengths:**

The paper shows two important results on the behavior of random feature models: first, that they can always improve with scale (both in terms of number of features and number of examples), and second, that overfitting might be necessary in some problem settings. To my understanding, these results unify previously studied settings, and present theoretical results that are more aligned with behaviors of large models in practice. The experiments seem to be convincing that the assumptions made in the analysis are reasonable in practical settings.

**Weaknesses:**

One point that I think the authors should clarify is in which cases the analysis is "rigorous", in a sense that the results follow from the problem assumptions, and in which cases the derivations are loose. It would be helpful, even for the "nonrigorous" steps, where some quantity is approximated, to state what are the precise bounds on the estimations. For the theorem statements in the paper, the way they are currently written, it is not clear if they truly follow from the assumptions, or are there some additional approximations (beyond the one covered in Assumption A) that are used in the analysis.

Additionally, for the results in Section 5, it is not clear what is the role that the noise plays in the analysis. Clearly, for large enough noise, overfitting hurts performance, i.e. - there is a predictor for which train error might be high (or even very high). While I understand that the noise might need to be scaled down (i.e., not analyze the setting of constant noise), it seems that the results in the paper only cover relatively low noise setting. What happens for higher noise levels? Do we expect overfitting to be unnecessary, or even hurt performance?

Minor:
Page 7, Target noise paragraph - "unlike is conventional" is probably a typo.

**Questions:**

See above

---

> ### Author Response · Authors · 2023-11-15
> **Author response**
>
> We thank the reviewer for their favorable evaluation! To the first comment re: rigor, while we are fully rigorous in all conclusions proven starting from eigenframeworks (i.e. Theorems, Lemmas, etc.), we opted for a streamlined, nonrigorous approach for our derivation of the RF eigenframework, adopting the philosophy typical in statistical physics where one makes justified approximations and then checks against numerics at the end of the calculation. As we discuss at various points, we expect our RF eigenframework to become exact in a proportional limit (e.g., precisely the limit taken by Bach et al (2023)). The analysis of Bach (2023) is rigorous under the Gaussian ansatz, and one can likely use the same techniques to obtain a rigorous proof for our RF eigenframework.
>
> Regarding the noise, our analysis applies even at high noise. Indeed, Figure 2 includes traces with relative noise variance as large as 50! We focus most of the exposition on the case of small noise because this is an interesting and realistic case.
>
> Thank you again for the review!

---

### Official Review · Reviewer_xXiV · 2023-11-01

**Soundness:** 4 excellent
**Presentation:** 4 excellent
**Contribution:** 2 fair
**Rating:** 6
**Confidence:** 4

**Summary:**

This work provides a unified formulation of the generalization error for random feature (RF) and kernel ridge regressions (KRR). While it is heuristically derived, it covers some expressions of the generalization error obtained in the previous work and gives us a quantitative understanding depending on the width (# of random features), sample size, teacher noise, and ridge penalty. This theoretical evaluation matches well empirical experiments. In addition, by focusing on the KRR limit, the authors succeeded in obtaining a more detailed evaluation for the case of power-law eigenstructure. That is, they provide an explicit formulation of the generalization error depending on the exponents of the student's and teacher's eigenstructures.

**Strengths:**

(i) While there are numerous individual analyses of RF and KRR under various conditions, this study attempts to provide a unified perspective. This endeavor, even if the derivation is heuristic, will be important for establishing a foundation for understanding.

(ii) While tackling such a significant (big) question, the study also provides technically solid contributions. In particular, this work gives novel continuum approximations for the generalization errors depending on the power-law eigenstructures (in particular, Lemma 1,10). This enables us to quantitatively understand the obligatory conditions for the "more is better" in the case of overfitting. This work also includes other technically solid and informative results such as remarks on the appropriate scaling of target noise (eq. 7), connection to the previous work (Section E), and the sophisticated evaluation of the power-law eigenstructures (Section B).

(iii) The paper is very well structured and articulated, with various efforts made to facilitate the reader's comprehension. This implies the author's deep understanding of RF and KRR.

**Weaknesses:**

(i) While the analysis for the power-law eigenstructure is solid and concrete, the other part seems not so surprising as the theory. I mean, the heuristic derivation is rather straightforward.  It seems that the authors just substituted a heuristically-derived terms for the kernel matrix of RF (eqs.26-30) into a well-known expression of the generalization error (eqs.13, 14). I understand that providing a unified formulation itself is a very curious and interesting attempt, though.  Please refer to my Question (i) for more details.



(ii) The definition of overfitting in this paper is unique, and its relation to the ridgeless case seems unclear. considering a ridgeless case seems appropriate when talking about overfitting to training samples. Indeed, Mei & Montanari 2019 have reported a certain condition where generalization error is minimized in a ridgeless setting and this was one of the main contributions.
In contrast, the current study does not directly analyze the ridgeless case but rather allows a non-zero fitting ratio in its analysis. For example, while Theorem 2 contemplates the optimal ridge, it's not explicitly stated when the optimal ridge becomes zero. I am also curious whether the noise condition in Corollary 1 works as a necessary and sufficient condition where the ridge-less case gives the best performance (I mean, the generalization error is minimized).

**Questions:**

**Major ones**

(i)  Regarding Weakness (i), eq.(13) seems a starting point to derive the omniscient risk estimate. However, the authors give not enough explanation on how and which paper has derived this equation.  As long as I know, Sollich, 2001, Bordelon et al., 2020 obtained this expression for KRR by using some heuristic derivations. Did the other work that you cited, i.e., Jacot et al., 2020a; Simon et al., 2021; Loureiro et al., 2021; Dobriban &Wager, 2018; Wu & Xu, 2020; Hastie et al., 2020; Richards et al., 2021,  explicitly give this expression (13) as well? If so, under what conditions does each study derive? Are they mathematically rigorous?  Because this equation is the starting point of all other analyses, it should be clearly and carefully described.

(ii) I am confused about what the authors try to claim in Figure 2. Do you intend to claim that overfitting is obligatory only for the noiseless case ($\sigma_{rel}^2=0$)? For $\sigma_{rel}^2>0$, the minimal test MSE is obtained for non-zero fitting ratios. This implies that the generalization is the best for non-overfitting situations and the overfitting seems *not* obligatory. This point is unclear.

**Minor ones**

(iii) In Section 5, the authors focused on the KRR limit (k -> infty). I am curious about at what point the finite-k case becomes hard to analyze.

(iv) Page 6: The definition of the fitting ratio seems to be wrong. It should not be R_tr /te:=E_te/E_tr but
would be E_tr/E_te.

---

> ### Author Response · Authors · 2023-11-15
> **Author response**
>
> We thank the reviewer for their favorable evaluation, noting in particular the unifying nature of our results. We will  respond to the two stated weaknesses.
>
> (i) We understand the reviewer’s comments to mean that, while they find our study of overfitting for powerlaw tasks (Section 5) solid and concrete, they find the RF eigenframework in Section 4 to be less surprising. This is fair: we are generally less interested in the RF eigenframework itself than in what new things we can conclude from it (namely the “more is better” result of Theorem 1). If some prior work had given a simple derivation and statement of this general result, we would have been happy just citing it. As nobody had done that to the best of our knowledge and it seems generally useful, we decided to. We also note that the simplicity with which a result may be derived is not necessarily indicative of the result’s importance, and the result reached here is indeed fairly general :)
>
> (ii) We understand that the reviewer finds unclear the relation of our “fitting-ratio-based” notion of overfitting to the simpler, more common approach of simply studying the ridgeless case (and perhaps comparing to the optimally-regularized case). To clear this up: the model is ridgeless ($\delta = 0$) precisely when the fitting ratio is zero ($\mathcal{R}\_{tr/te} = 0$). When we find that $\mathcal{R}_{tr/te} = 0$ is optimal, we are precisely finding that zero ridge is optimal. We realize that we missed saying this explicitly, and so we have added some appropriate text where we introduce the fitting ratio. The reviewer’s interpretation of Corollary 1 is correct (and indeed answers the reviewer’s concern regarding Theorem 2), and the text will say so.
>
> To the reviewer’s questions:
>
> (Qi): All the cited works derive the same eigenframework. Right now, the best (i.e. most general while still rigorous) source for this eigenframework is probably [Cheng and Montanari (2022)](https://arxiv.org/abs/2210.08571), and this could be taken as the canonical starting point if one is required. We have edited the text to say this.
>
> (Qii): Yes, this is the intended claim. The most important curve is the one with zero noise, because this is the true, unmodified task. The point of adding noise is that Corollary 1 predicts that adding noise should make overfitting no longer obligatory, and we are testing that. Comparing to noisy curves is also a way of stressing that the unmodified tasks seem to not have significant noise.
>
> (Qiii): This is a good question. The finite-k case is algebraically harder to handle: from the start, one has the two coupled equations (2) defining implicit constants instead of just one. To undertake a similar powerlaw analysis, one might write test error in terms of $\gamma$, then write $\gamma$ in terms of the train-test ratio, and finally optimize over the train-test ratio. The test error in the finite-k case would be significantly more complicated than in the infinite-k case (Lemmas 4 and 10), but this might work. It may be a doable thing for a future paper.
>
> (Qiv): Thanks for catching this. We’ve fixed it.
>
> We thank the reviewer again for their review. If they feel we have addressed some of their concerns, we ask that they consider adjusting their score accordingly.

---

> > ### Comment · Reviewer_xXiV · 2023-11-23
> > **Reply to Authors' respose**
> >
> > Thank you for your kind reply. I understand from your reply that there is no misunderstanding regarding my comprehension, so I have decided to keep the score as it is.
> >
> > My evaluation is that, while it is nice to provide a unified perspective and concrete results based on the eigen spectrum, the 'more is better' result itself is not surprising. As the authors mention,
> >
> >  >adding noise should make overfitting no longer obligatory.
> >
> >  In cases with less noise, we can easily expect that  'more is better' is obligatory if we consider specific scalings. For instance, Canatar et al. 2021 demonstrated that for polynomial scaling, i.e., sample size = (input dimension)^(natural number)), the learning curve monotonically decreases and 'more is better' holds under specific small (or zero) noises as characterized in their phase diagram.

---

### Meta-Review · Area_Chair_cHyH · 2023-12-11

**Metareview:**

The paper makes a solid contribution to the theory of deep learning, specifically, towards a deeper understanding of the benefits of overparameterization. The authors essentially study kernel and random features regression and rigorously establish that infinite overparameterization (kernel regression) is always preferable to finite width regression (random features). They similarly demonstrate the need for more training by showing that ridge regularization should essentially vanish for achieving optimal test risk (under suitable but practically-meaningful spectral conditions). Here ridge regularization provides an inverse-proxy for the amount of computation (e.g. number of epochs). Overall, I believe this will make a nice addition to the ICLR venue.

**Justification For Why Not Higher Score:**

It can be bumped up but main results may not be that surprising given our current knowledge on overparameterization.

**Justification For Why Not Lower Score:**

Reviewers and AC are unanimous regarding acceptance.

---

### Decision · Program_Chairs · 2024-01-16

Accept (poster)